# Hyperacetylated chromatin domains mark cell type-specific genes and suggest distinct modes of enhancer function

Sierra Fox [1], Jacquelyn A. Myers[2,3], Christina Davidson [2], Michael Getman [2], Paul D. Kingsley [2], Nicholas Frankiewicz [2] & Michael Bulger[2 ✉]

Stratification of enhancers by signal strength in ChIP-seq assays has resulted in the establishment of super-enhancers as a widespread and useful tool for identifying cell type-specific, highly expressed genes and associated pathways. We examine a distinct method of stratification that focuses on peak breadth, termed hyperacetylated chromatin domains (HCDs), which classifies broad regions exhibiting histone modifications associated with gene activation. We find that this analysis serves to identify genes that are both more highly expressed and more closely aligned to cell identity than super-enhancer analysis does using multiple data sets. Moreover, genetic manipulations of selected gene loci suggest that some enhancers located within HCDs work at least in part via a distinct mechanism involving the modulation of histone modifications across domains and that this activity can be imported into a heterologous gene locus. In addition, such genetic dissection reveals that the super-enhancer concept can obscure important functions of constituent elements.

[1] Department of Biochemistry and Biophysics, University of Rochester Medical Center, Rochester, NY 14642, USA. [2] Center for Pediatric Biomedical Research, Department of Pediatrics, University of Rochester Medical Center, Rochester, NY 14642, USA. [3] Genomics Research Center, University of Rochester Medical Center, Rochester, NY 14642, USA. ✉email: Michael_Bulger@URMC.Rochester.edu

Enhancers are *cis*-regulatory DNA sequences that are bound by transcriptional activators that regulate gene promoters[1–5]. They are also historically characterized by their ability to function over distances of anywhere from 100 bp to more than 1 Mb. Whole-genome methods of identifying enhancers rely on associated histone modifications, common transcriptional cofactors, and/or cell type-specific transcription factors (TFs). These methods suggest that mammalian genomes harbor a large number of enhancer sequences — perhaps 1–2 million — and that they represent the most numerous and significant *cis*-acting sequence determinants of cell-type-specific gene expression.

Fundamental issues regarding enhancer function, however, remain unclear. For example, the dominant model for how enhancers communicate with their cognate gene promoters, termed looping, involves direct interactions between factors bound to enhancers and factors bound near promoters. Evidence for such interactions, however, has provided little insight into how a distal enhancer finds a gene promoter, or how it distinguishes among potential promoters in gene-dense regions. Moreover, some evidence suggests that mechanisms of enhancer-promoter communication may be more varied[3,5].

Additional insight into enhancer function can be obtained by discerning functional differences between them. One attempt to classify enhancers according to strength, as determined by the signal intensity in ChIP-seq assays for associated enhancer marks, has led to the classification of super-enhancers, which have been shown to be associated with highly expressed genes that define cell identity[6–8]. Obvious functional differences have not emerged from studies of super-enhancers, however, with the only reported distinction being a higher diversity and/or number of TF binding sites mapping to super-enhancers[9,10]. Mechanistically, there is as yet no indication that, aside from the strength of activation, these sequences are intrinsically different from other enhancers that exhibit weaker signals for enhancer-defining marks.

We have previously investigated enhancer-associated histone modifications from the perspective of peak breadth, as opposed to signal strength. We characterized specific regions as hyperacetylated chromatin domains (HCDs), defined as continuous genomic regions exhibiting significant enrichment for histone hyperacetylation and other marks associated with active transcription, such as H3K4 dimethylation (H3K4Me2). From this analysis, we identified a novel enhancer within the murine β-globin locus, termed HS-E1, which is required for the formation of such a broad region of histone hyperacetylation encompassing the two genes within the cluster that are expressed during primitive erythropoiesis[11,12]. Our results suggested a distinction between enhancers that mediate broadly distributed changes in chromatin structure vs enhancers that work via other mechanisms.

To further investigate this, we perform ChIP-seq analyses of specific histone modifications in primary murine erythroid, retinal, and intestinal epithelial cells and rank peaks according to breadth. We find that an HCD ranking that utilizes a combination of two histone marks suffices to identify a subpopulation of genes that are both more highly expressed and more cell type-specific than those associated with super-enhancers. Moreover, the deletion of enhancers found in loci with or without an HCD identifies functional differences between enhancers that have the ability to modulate long-range chromatin structure and those that do not. Insertion of an HCD-associated enhancer into another locus results in the formation of an HCD, further suggesting a distinct function for this class of enhancer.

## Results

### HCDs mark highly expressed, erythroid-specific genes. Our prior studies suggested that HCDs at specific genomic regions,

such as the murine β-globin locus, are controlled by enhancers[12]. In an effort to define the genome-wide distribution of such domains, we analyzed (1) ChIP-seqs we performed using e14.5 murine fetal liver, which is comprised of 70–80% erythroid cells, using antibodies specific for H3K27 acetylation (H3K27Ac), and for H3K4 mono-, di- and trimethylation (H3K4Me1, H3K4Me2, H3K4Me3); (2) ATAC-seq[13] we performed using sorted proerythroblasts from e14.5 murine fetal liver; and (3) publicly available ChIP-seq data sets from e14.5 murine fetal liver for the erythroid TFs GATA-1 and SCL/Tal1 (ref. [14]) (Fig. 1).

We identified HCDs using our H3K27Ac and H3K4Me2 ChIP-seq data sets derived from e14.5 murine fetal liver by ranking MACS2 (ref. [15]) peaks by breadth. The establishment of a formal definition for a domain requires the application of a subjective cutoff value; for our purposes, rather than an absolute peak breadth we chose the top 2% of MACS2 peaks ranked by breadth. We did this because we found that a cutoff based on ranking, as opposed to absolute peak breadth, translated more consistently between different data sets. Our other criterion was that this produced a list of HCDs that was comparable in size to the list of super-enhancers called from the same data sets (see below), and thus facilitated a comparison of the two methods.

To arrive at this list of HCDs, we first intersected the replicates of our ChIP-seqs for each histone modification (H3K27Ac or H3K4Me2), and then applied the 2% cutoff to each intersection, resulting in 420 H3K27Ac and 760 H3K4Me2 peaks. We then identified the H3K27Ac and H3K4Me2 peaks that overlapped within the genome and merged them by taking their union. This resulted in a final tally of 216 regions we term hyperacetylated chromatin domains (HCDs) in primary murine fetal liver (Supplementary Fig. 1a, Supplementary Data 1).

When choosing criteria to call super-enhancers, we attempted to stay as true to the original method as possible, while ensuring the data sets were comparable to our hyperacetylated domains. We, therefore, used lineage-specific TF (GATA-1 and SCL/Tal1) peaks to identify a set of enhancers in fetal liver[14]. We then ranked these enhancers based on signal intensity in our H3K27Ac and H3K4Me2 ChIP-seqs, since these are the marks that we used to identify hyperacetylated domains. We used the ROSE algorithm[6,7] with the default stitching distance (12.5 kb) and a ±500 bp TSS exclusion zone, which resulted in 307 super-enhancers ranked on H3K27Ac and 214 on H3K4Me2. To identify super-enhancers for both H3K27Ac and H3K4Me2 we took the union of the 307 and 214 super-enhancers, only where these regions overlapped, for a final tally of 173 murine erythroid super-enhancers. (Supplementary Fig. 1b, Supplementary Data 1). Examples of loci that exhibit HCDs, super-enhancers, both features, or neither are shown in Fig. 1.

Insofar as signal strength is used as a proxy for enhancer strength, the fundamental utility of super-enhancer identification has been the ability to identify associated genes that encode factors important for cell-type specificity. We therefore compared the properties of genes associated with HCDs to those associated with super-enhancers. For the association of genes with super-enhancers, the most commonly used method is the nearest neighbor (i.e., closest gene), and so we used this method to associate super-enhancers with genes. We used the same approximation for HCDs, although for 205 out of the 216 regions we have classified as HCDs in murine fetal liver, the nearest active genes are actually located within them. The remaining 11 HCDs harbor peaks of H3K4Me3 (Supplementary Data 1), suggesting the presence of unannotated gene promoters, but for the sake of consistency, we still associated these HCDs with the nearest annotated genes. Using a database of Affymetrix-derived gene expression in murine fetal liver (ErythronDB)[16,17], we then compared expression levels for genes associated with

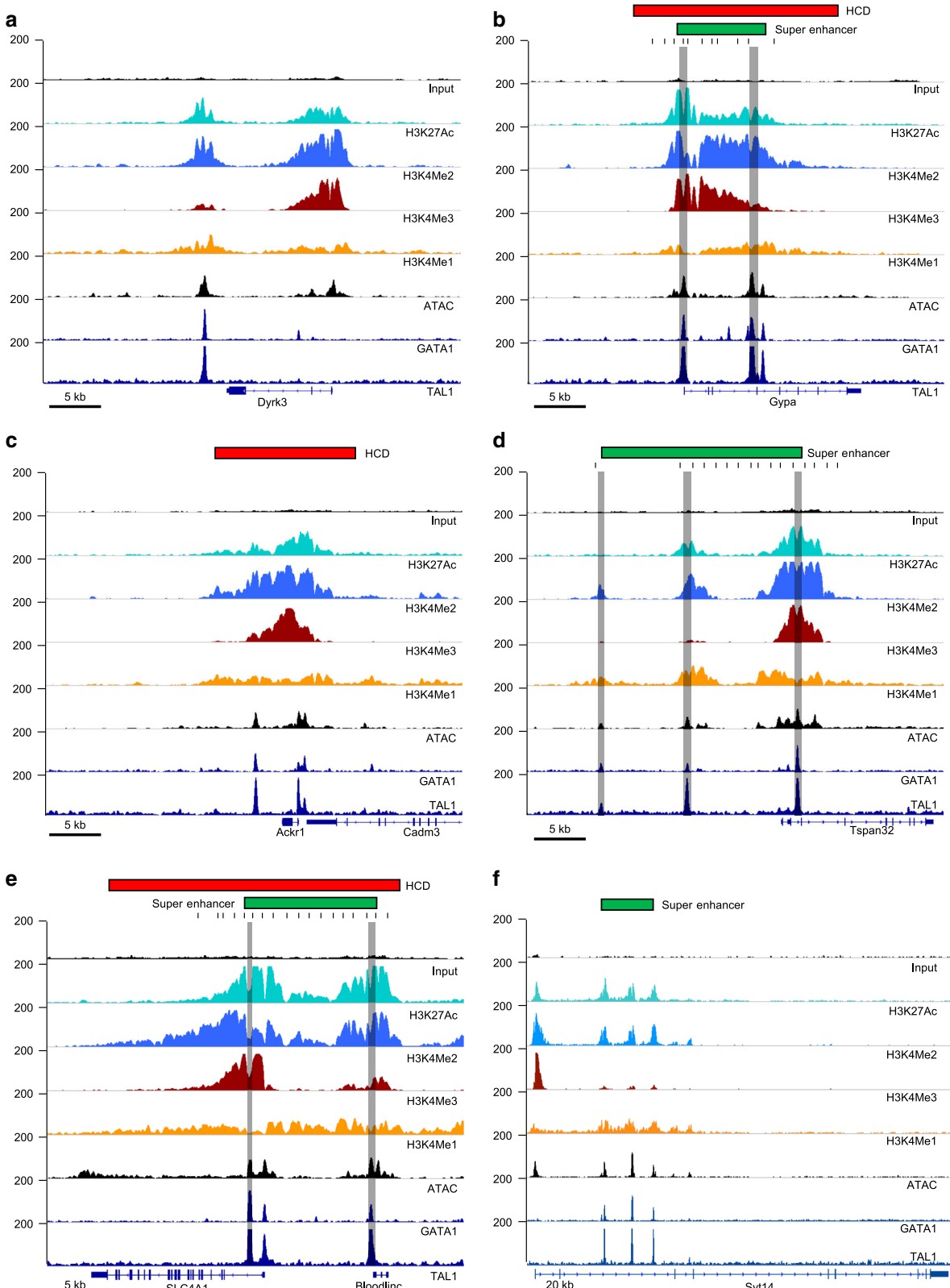

**Fig. 1 ChIP-seq profiles at selected gene loci.** Tracks show read densities for the indicated histone modifications or transcription factors, with additional tracks for the Input control at top and for ATAC-seq. Genes and scale are shown at the bottom, and peak calls for super-enhancers and/or hyperacetylated chromatin domains (HCDs) at the top. Gray shading indicates regions tested by deletion in MEL cells by CRISPR/Cas9-mediated genome editing. Black bars indicate regions amplified in ChIP-qPCR experiments. **a** The Dyrk3 locus, harboring a putative enhancer that is called neither a super-enhancer nor an HCD. **b** The *Gypa* locus, harboring both a super-enhancer and an HCD. **c** The Ackr1/Cadm3 locus, harboring an HCD but not a super-enhancer. **d** The *Tspan32* locus, harboring a super-enhancer but not an HCD. **e** The *SLC4A1/Bloodlinc* locus, harboring an HCD and a super-enhancer. **f** The Syt14 locus, harboring a super-enhancer but not an HCD.

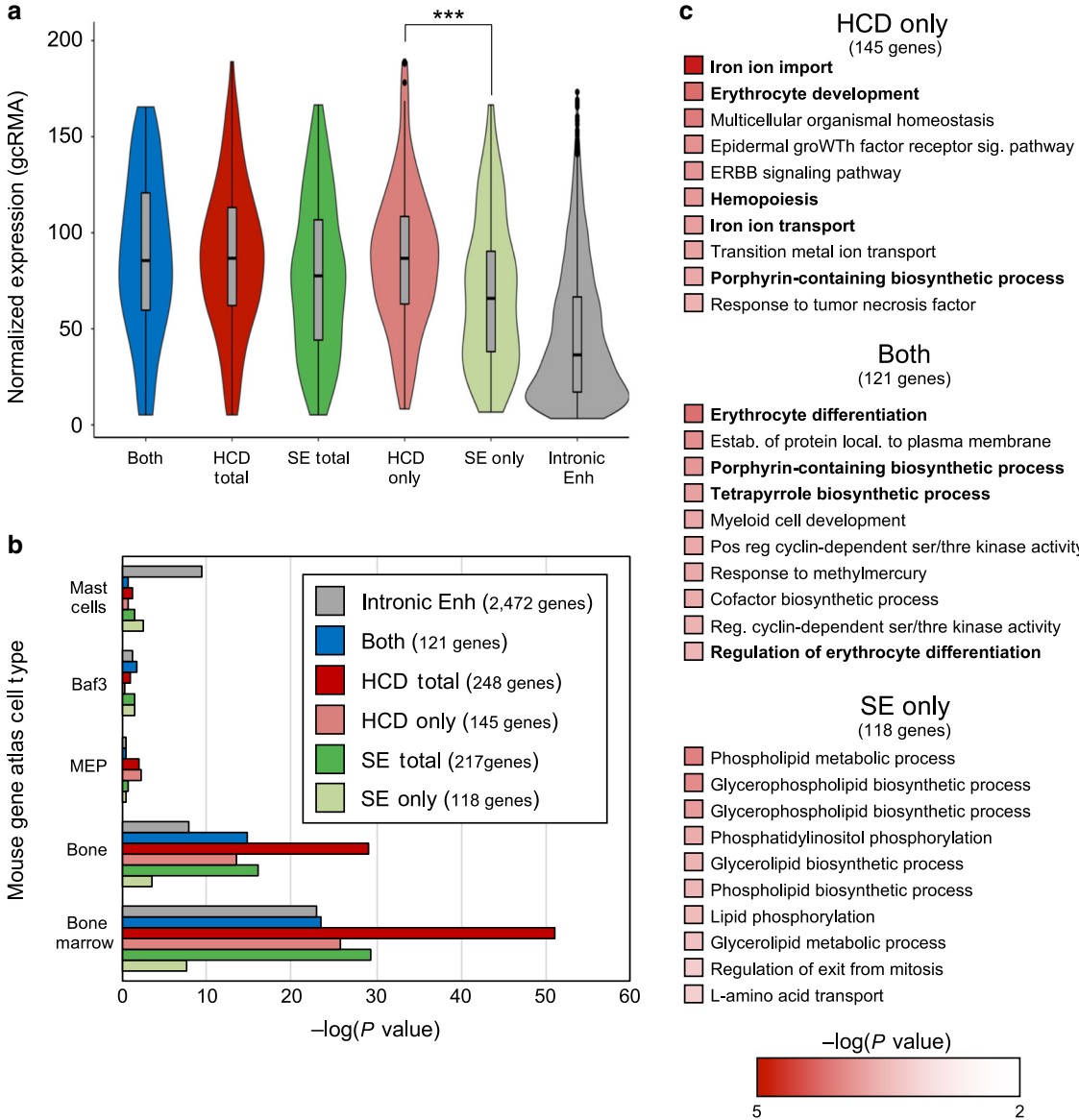

**Fig. 2 Comparison of genes associated with super-enhancers vs HCDs in murine erythroid cells. a** Violin plots of expression of genes associated with HCDs, super-enhancers or both. In box plots, the center line represents the median, the box limits represent the 25th and 75th percentiles and the whiskers represent 1.5 times the interquartile range. *P*-values were calculated with a two-tailed Student's *t*-test, *P* = 0.00056. Expression for genes associated with all putative enhancers located within introns is shown for comparison. **b** Bar graph showing *P*-value for Enrichr cell-type enrichment for the 5 cell types with the highest scores for each category. **c** Listings of the top ten GO terms for biological processes for the indicated groups; erythroid-specific terms are in boldface type.

HCDs and genes associated with super-enhancers (Fig. 2a). We find that HCD-associated genes are expressed at significantly higher levels in murine fetal liver than those associated with super-enhancers.

Not only did we find that HCDs were associated with more highly expressed genes than super-enhancers, but we also found that these genes are better able to identify erythroid cell types in the Mouse Gene Atlas than genes associated with super-enhancers, using Enrichr cell type analysis software[18,19] (Fig. 2b, Supplementary Data 2). Functional enrichment analysis, also through Enrichr, shows that genes associated with HCDs are more likely to identify terms specific for erythroid cells. Seven of the ten most enriched terms for genes associated solely with HCDs are specific for erythroid biology, while in contrast, only two are erythroid-specific using genes associated solely with super-enhancers (Fig. 2c, Supplementary Data 3). Notably, genes

found to be associated with both an HCD and a super-enhancer exhibit expression levels similar to those associated with an HCD alone, and thus also higher than those associated only with super-enhancers. In Enrichr analysis, this population of genes also results in the identification of terms more obviously applicable to erythroid biology than with genes associated solely with super-enhancers.

**HCDs vs super-enhancers in other cell types and in human cells.** To determine how generalizable our comparison of HCDs and super-enhancers is, we applied these analyses to a selection of publicly available data sets. We were able to perform domain analysis and super-enhancer analysis on ChIP-seqs derived from human primary cultured CD34[+] erythroid cells[20] as with our murine fetal liver-derived data sets (Fig. 3, Supplementary Fig. 2,

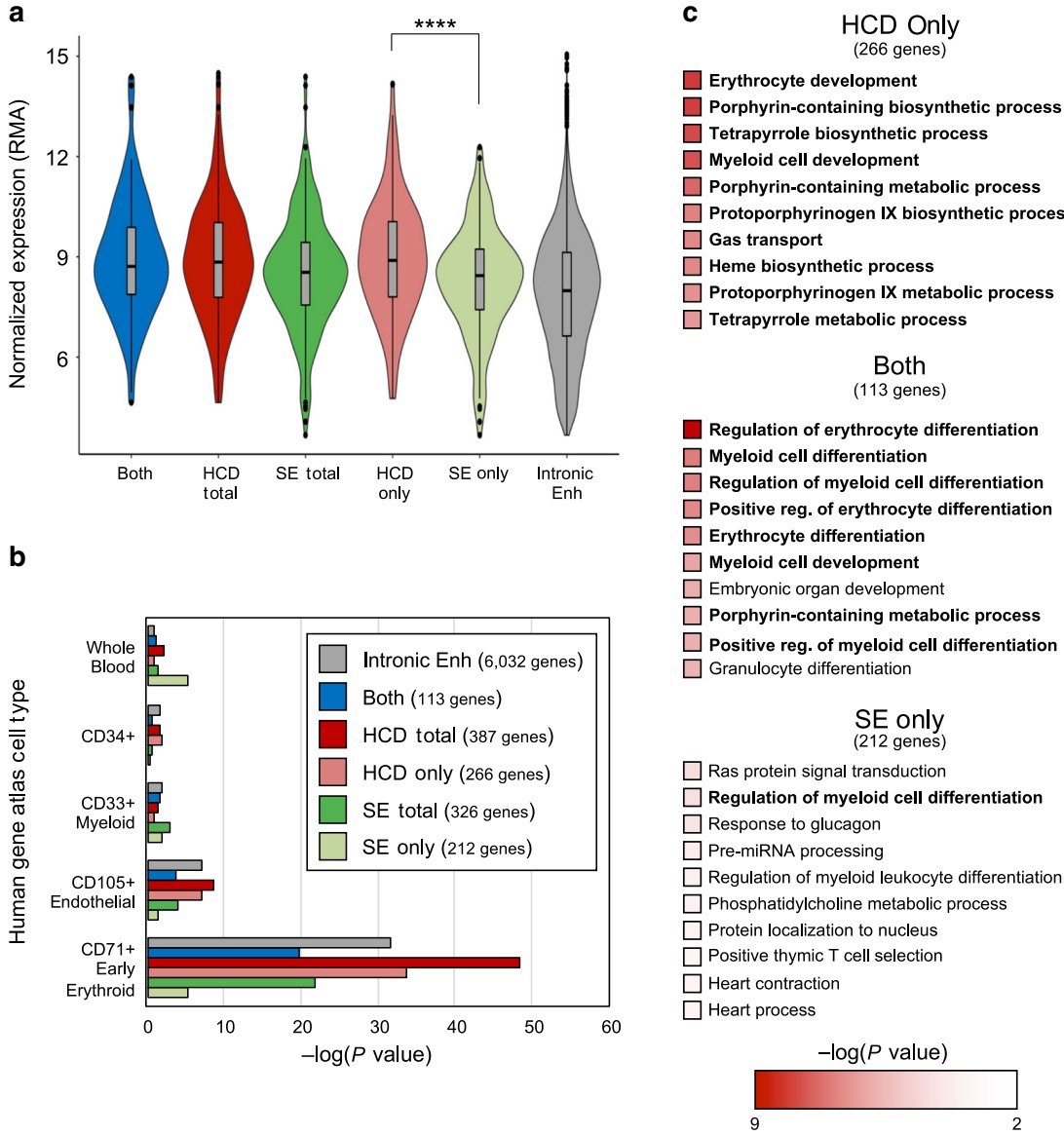

**Fig. 3 Comparison of genes associated with super-enhancers vs HCDs in human erythroid cells. a** Violin plots of expression of genes associated with HCDs, super-enhancer, or both. In box plots, the center line represents the median, the box limits represent the 25th and 75th percentiles and the whiskers represent 1.5 times the interquartile range. *P*-values were calculated with a two-tailed Student's *t*-test, *P* = 0.00073. Expression for genes associated with all putative enhancers located within introns is shown for comparison. **b** Bar graph showing *P*-value for Enrichr cell-type enrichment for the 5 cell types with the highest scores for each category. **c** Listings of the top ten GO terms for biological processes for the indicated groups; erythroid-specific terms are in boldface type.

Supplementary Data 4). This analysis produced 302 HCDs and 326 super-enhancers, and the same trend that we see in mouse erythroid cells holds true in human erythroid cells: genes associated with HCDs are both more highly expressed than genes associated with super-enhancers, (Fig. 3a, Supplementary Data 5) and appear to be more erythroid-specific (Fig. 3b, c, Supplementary Data 6).

We performed additional comparisons of HCD and super-enhancer analyses using available data sets derived from intestinal epithelium[21] (Supplementary Fig. 3, Supplementary Data 7) and retina[22,23] (Supplementary Fig. 4, Supplementary Data 8). In these cases, while all rankings are still performed using H3K27Ac and H3K4Me2, enhancer calls for super-enhancer analysis are derived from ATAC-seq instead of TFs, and intestinal epithelial domains were reduced to the top 1% of H3K27Ac/H3K4Me2 peaks ranked by breadth, again to arrive at a list of HCDs

comparable in size to the list of super-enhancers (see Methods). In both cases, the HCDs identify more highly expressed genes (Supplementary Figs. 5A, 6A), and genes more closely associated with the specific tissue type, than do super-enhancers (Supplementary Figs. 5B, C, 6B, C, Supplementary Data 9–12).

**HCDs can be identified using multiple chromatin signatures**. Stratification of ChIP-seq-derived H3K4Me3 peaks by breadth has similarly been demonstrated as a useful tool for the identification of highly expressed genes important in determining cell identity[24]. We, therefore, performed a comparison of genes associated with HCDs to genes associated with the broadest 5% of H3K4Me3 peaks in our murine erythroid ChIP-seqs (Supplementary Fig. 7, Supplementary Data 13–15). In this comparison, HCDs identified a population of genes that was more highly

expressed than that associated with the broadest H3K4Me3 peaks, but the difference was not as great as between HCDs and SEs. Enrichr analysis, however, indicates that H3K4Me3 peaks identify a population of genes more closely associated with erythroid cell types in the Mouse Gene Atlas, along with closer association with erythroid-specific terms in functional enrichment analysis. The set of genes within the broadest 5% of H3K4Me3 peaks, however, was considerably larger than that associated with HCDs, with fully 94% of HCD-associated genes falling within this population. Thus, this analysis represents a comparison between the set of HCD-associated genes and a larger set that subsumes HCDs.

To assess the suitability of other combinations of histone marks for HCD analysis, we identified the set of HCDs defined using the H3K27Ac mark alone, and the set defined by the combination of H3K27Ac and H3K4Me3, using our murine fetal liver and the murine retina ChIP-seq data sets (Supplementary Figs. 8 and 9). We find that the population of genes associated with HCDs defined by H3K27Ac and H3K4Me3 is somewhat more highly expressed than that associated with our H3K27Ac/H3K4Me2 HCDs, and are similarly associated with cell type-specific terms in functional enrichment analysis. These genes, however, are less closely associated with appropriate cell types in the Mouse Gene Atlas, and in fact, fail to outperform SEs in this measure. In contrast, limiting the definition of HCDs to the H3K27Ac mark alone results in a decrease in expression of associated genes, to levels similar to those observed with SE-associated genes. Measurements of cell-type specificity, however, are nearly the same as with H3K27Ac/H3K4Me2-defined HCDs, and similarly, outperform SE-associated genes. Taken together, the results unsurprisingly show that deriving HCDs with different activation-associated histone marks can produce substantially overlapping but distinct populations of associated genes, but that in general these are still more highly expressed and/or specific than SE-associated genes.

**HCD formation at selected gene loci is enhancer-dependent.** Our previous characterization of HS-E1 within the β-globin locus as an enhancer required for the formation of an HCD implied that classification of enhancers according to peak breadth could distinguish between enhancers with different functions[12]. To investigate this, we performed CRISPR/Cas9-mediated genetic manipulation of selected gene loci in murine erythroleukemia (MEL) cells[25,26]. These are transformed cells that exhibit a phenotype similar to proerythroblasts, which can be induced to mature (but not enucleate) by incubation in 2% DMSO. Maturation is associated with cessation of cell division in a majority (>90%) of cells and dramatic upregulation of erythroid-specific genes.

As an example locus exhibiting an HCD, we chose the region harboring the gene encoding glycophorin A (*Gypa*), a late-stage erythroid cell surface marker that is highly upregulated upon MEL cell maturation. The region exhibits a domain that encompasses the gene promoter and a pair of putative enhancers within the third and fourth introns of the gene (Fig. 1b). These enhancers, together with the region near the gene promoter, are also called as a super-enhancer by ROSE-based analysis. Using a CRISPR/Cas9-based strategy we deleted a 700 bp region encompassing the major GATA1 binding site within intron 3, while leaving exon 4 and the splice acceptor intact.

We measured the expression of *Gypa* in differentiating (DMSO-treated) MEL cells, which more closely resemble e14.5 fetal liver than undifferentiated MEL cells, and saw that *Gypa* expression decreased >10-fold in cell lines harboring the deletion (Fig. 4a). ChIP-qPCR analysis of the *Gypa* locus indicates that the deletion of the enhancer has a significant effect on levels of

H3K4Me2 (Fig. 4b) and H3K27Ac (Fig. 4c) across the entire region, consistent with a requirement for the enhancer in HCD formation.

Given that the *Gypa* HCD occurs largely within the transcribed region of the gene, however, the effect of enhancer deletion on histone modifications could represent a secondary consequence of decreased transcription. To address this, we created MEL cells harboring a deletion of the *Gypa* gene promoter. These cells exhibit negligible expression of *Gypa* (Fig. 4a), but levels of histone modifications are not affected to nearly the same degree as with the enhancer deletion (Fig. 4b, c). The results suggest that the histone modifications that define the *Gypa* HCD are largely a direct consequence of enhancer activity.

Another HCD occurs within the *SLC4A1* locus, harboring the gene that encodes Band 3, an erythroid cell membrane anion transporter. The HCD encompasses a portion of the transcribed region along with sequences extending ~12 kb upstream of the gene promoter, including a lncRNA gene, *Bloodlinc*, that has been reported to be transcribed in erythroid cells[27]. While we detected the expression of this lncRNA in our murine fetal liver samples, we do not detect expression in MEL cells. We can identify putative enhancers, as indicated by TF binding, at locations +1.1 kb and −10.4 kb from the *SLC4A1* transcription start site (TSS); the latter is located near the 3′ end of the annotated *Bloodlinc* gene.

We evaluated the effects of separate CRISPR/Cas9-mediated deletions on these enhancers. *SLC4A1* is not expressed at measurable levels in proliferating MEL cells, so we confined our analyses to differentiating MEL. Interestingly, individual deletions of either enhancer result in reductions in *SLC4A1* mRNA levels (Supplementary Fig. 10A), indicating that both enhancers are necessary for normal gene expression. Moreover, both deletions result in decreases in enrichments for H3K27Ac and H3K4Me2 locus-wide (Supplementary Fig. 10B, C), with two exceptions: (1) H3K4Me2 levels proximal to the *SLC4A1* promoter are minimally affected; (2) the +1.1 kb enhancer deletion has no significant effect on these modifications in the vicinity (±1 kb) of the −10.4 enhancer. Thus, at this locus, the HCD requires the activity of two enhancers, neither of which is sufficient for domain formation in the absence of the other. As with the *Gypa* locus, however, the HCD appears to be a consequence of enhancer activity and is associated with high-level gene expression.

**SEs not associated with an HCD do not affect histone signatures.** As an example locus exhibiting a super-enhancer, but not an HCD, we investigated the region harboring the gene encoding *Tspan32*, a member of the tetraspanin superfamily. The super-enhancer at this locus includes three putative elements, marked by GATA1 binding, located at −17, −10, and +0.7 kb relative to the TSS. We created deletions of each enhancer individually by CRISPR/Cas9-mediated genome editing in MEL cells. Notably, measurement of *Tspan32* gene expression in differentiating MEL cells indicated that only deletion of the +0.7 enhancer had a significant effect (Fig. 5a), while deletions of the other two enhancers showed no requirement in the maintenance of normal steady-state *Tspan32* mRNA levels. Moreover, none of the deletions, including the +0.7 enhancer, showed any effect on histone modifications (H3K27Ac or H3K4Me2) in the region encompassing the *Tspan32* gene promoter (Fig. 5b, c).

To further investigate this, we considered the possibility that the nearest neighbor assumption — e.g., that the super-enhancer or individual elements within it regulates only the *Tspan32* gene — might not be valid, and so we examined additional genes neighboring *Tspan32*. Hi-C data from murine cells[28] indicate that *Tspan32* resides within a topologically associating domain (TAD)

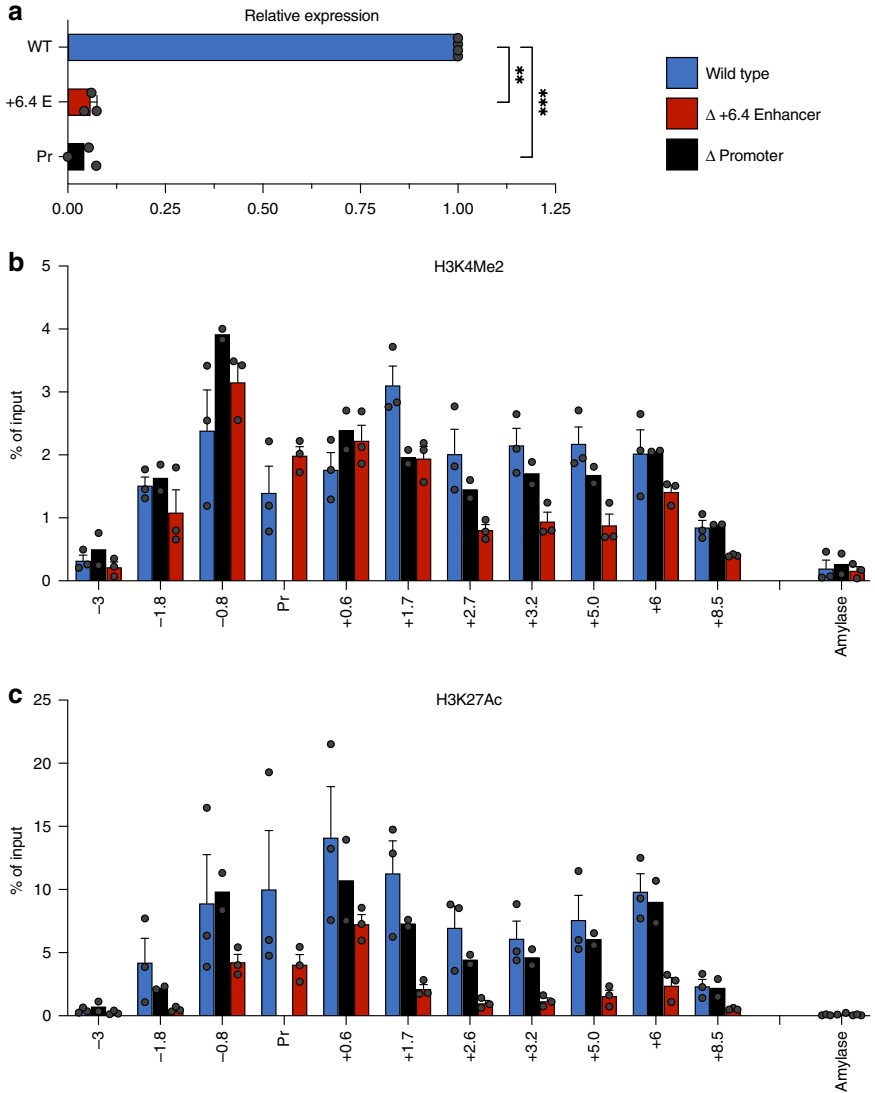

**Fig. 4 Deletions of the gene promoter and an enhancer in the *Gypa* locus in differentiating MEL cells. a** Bar graph showing the results of quantitative real-time PCR analysis of *Gypa* expression in wt MEL cells (MEL), cells harboring a deletion of the putative intronic enhancer region (6.4 Enh) or the *Gypa* gene promoter (Promoter). *P*-values were calculated using a two-tailed Student's *t*-test; **$P < 0.005$, ***$P < 0.0005$. **b**, **c** Bar graph showing the percent of input control obtained using PCR probes at the indicated locations (in kb) relative to the transcription start site for the *Gypa* gene in ChIP assays using antibodies specific for H3K4Me2 (**b**) or H3K27Ac (**c**). Amylase indicates a control probe within the inactive amylase gene locus. Results of all panels are means + s.e.m of at least three independent experiments for WT and Δ +6.4 Enh and two independent experiments for Δ Pr. Each colored circle represents the average of three technical replicates of an independent single-cell-derived homozygous enhancer knockout clone.

that contains two other genes expressed in erythroid cells, *Cd81* (encoding another tetraspanin superfamily member) and *Tssc4* (encoding a protein of unknown function), located 48 and 64 kb from the *Tspan32* TSS, respectively. Surprisingly, the deletion of the +0.7 enhancer had significant effects on the expression of both genes (Fig. 5a), which paralleled the effects we observed on the *Tspan32* gene. Thus, the +0.7 enhancer is required for normal expression of multiple neighboring genes in differentiating MEL cells.

We then utilized a publicly available transcriptomic database (ErythonDB)[16,17] to examine the expression of *Tspan32*, *Cd81,* and *Tssc4* during normal erythropoiesis. This indicated that *Tspan32* is upregulated later in erythroid maturation, while *Cd81* and *Tssc4* are downregulated. We, therefore, examined gene expression in our various MEL cell lines prior to differentiation, as an approximation of an earlier stage of erythroid maturation than that modeled by DMSO-treated MEL cells. Interestingly, in this environment, normal expression levels of all three genes

required the −10 kb element, while the other enhancers had smaller effects (Fig. 6). This suggests that the components of the *Tspan32* super-enhancer have differential activities at different stages of erythropoiesis, with a switch to dependence on the +0.7 kb element from the −10 kb element as maturation proceeds. Notably, ChIP-qPCR analysis of the promoter region of *Tspan32* indicates that, as in differentiating MEL cells, none of the enhancer deletions affects histone modifications in undifferentiated MEL cells (Fig. 6). This suggests that the specific modifications we have measured over the *Tspan32* promoter region are not sufficient for high-level gene expression, and that the *Tspan32* enhancers, at distinct maturational stages, function by a different mechanism that does not involve long-range modification of chromatin.

**Formation of an HCD by insertion of an enhancer at a heterologous locus.** Genetic manipulations in MEL cells indicate

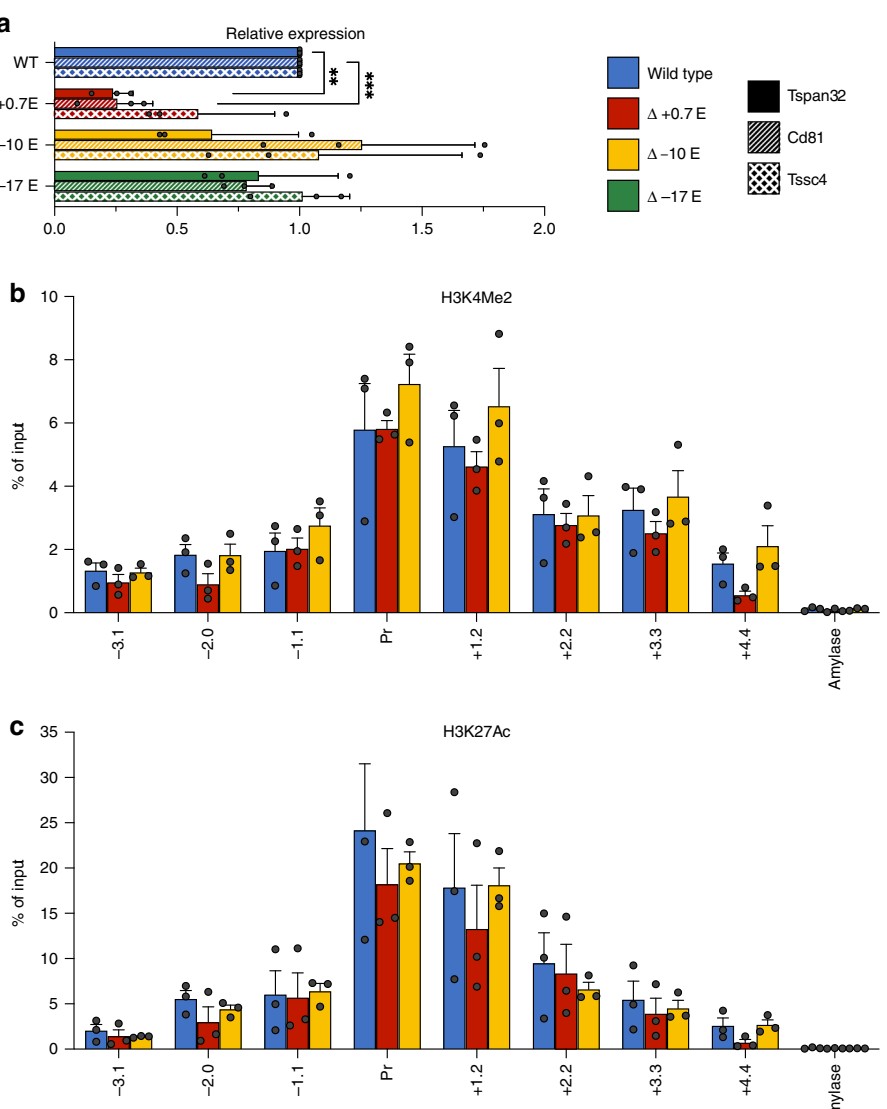

**Fig. 5 Effects of deletions of putative enhancers in the *Tspan32* gene locus in differentiating MEL cells. a** Normalized gene expression for *Tspan32*, *Cd81* and *Tssc4* measured by qrt-PCR in wt (MEL) and upon deletion of the indicated enhancer regions (0.7, −10, −17). *P*-values were calculated using a two-tailed Student's *t*-test; *$P < 0.05$, **$P < 0.005$, ***$P < 0.0005$. **b**, **c** Bar graphs showing % of input control for the indicated probes derived from qrt-PCR analysis of ChIPs using antibodies specific for H3K4Me2 (**b**) or H3K27Ac (**c**). Amylase indicates a control probe within the inactive amylase gene locus. Results of all panels are means + s.e.m of at least three independent experiments. Each colored circle represents the average of three technical replicates of an independent single-cell-derived homozygous enhancer knockout clone.

that HCD formation at the *Gypa* and *SLC4A1* gene loci requires the activity of specific enhancers within them. We, therefore, asked if the *Gypa* enhancer was also sufficient to form an HCD. To do this, we employed a CRISPR-in strategy, in which a portion of the *Gypa* enhancer was inserted into the location of the −10 kb element at the *Tspan32* locus. This was accomplished using a repair template consisting of the *Gypa* enhancer sequence flanked by homology arms, targeted to the site of the deleted *Tspan32* −10 kb element. We find that in subclones homozygous for this insertion, expression of the *Tspan32* gene increases in differentiating MEL cells (Fig. 7), where the *Gypa* enhancer is most active. Notably, the +0.7 kb element, which is also active in differentiating MEL, is still present in these cells, and so this represents super-activation over normal transcription levels in this context.

We also find, however, that introduction of the *Gypa* enhancer results in significant increases in H3K27Ac and H3K4Me2 at the site of insertion and for substantial distances in either direction,

including probes located 5–10 kb upstream. Thus, the *Gypa* enhancer is sufficient to induce the formation of an HCD when introduced into the *Tspan32* locus.

## Discussion

Ranking of enhancers by strength as determined by ChIP-seq signal intensity is now a common technique for meta-analysis of epigenomic data, with applications in revealing cell type-specific pathways and transcriptional regulatory networks. In this regard, the super-enhancer model has generated much interest, especially insofar as it can be applied to dysfunctional or disease states. In this study, however, we demonstrate that for the purposes of identifying the genes and pathways most important for cell phenotype, a ranking according to peak breadth of regions identified by ChIP-seq using antibodies for two commonly assayed histone modifications is more useful. The combination of H3K4Me2 and H3K27Ac domains — both of which are detected

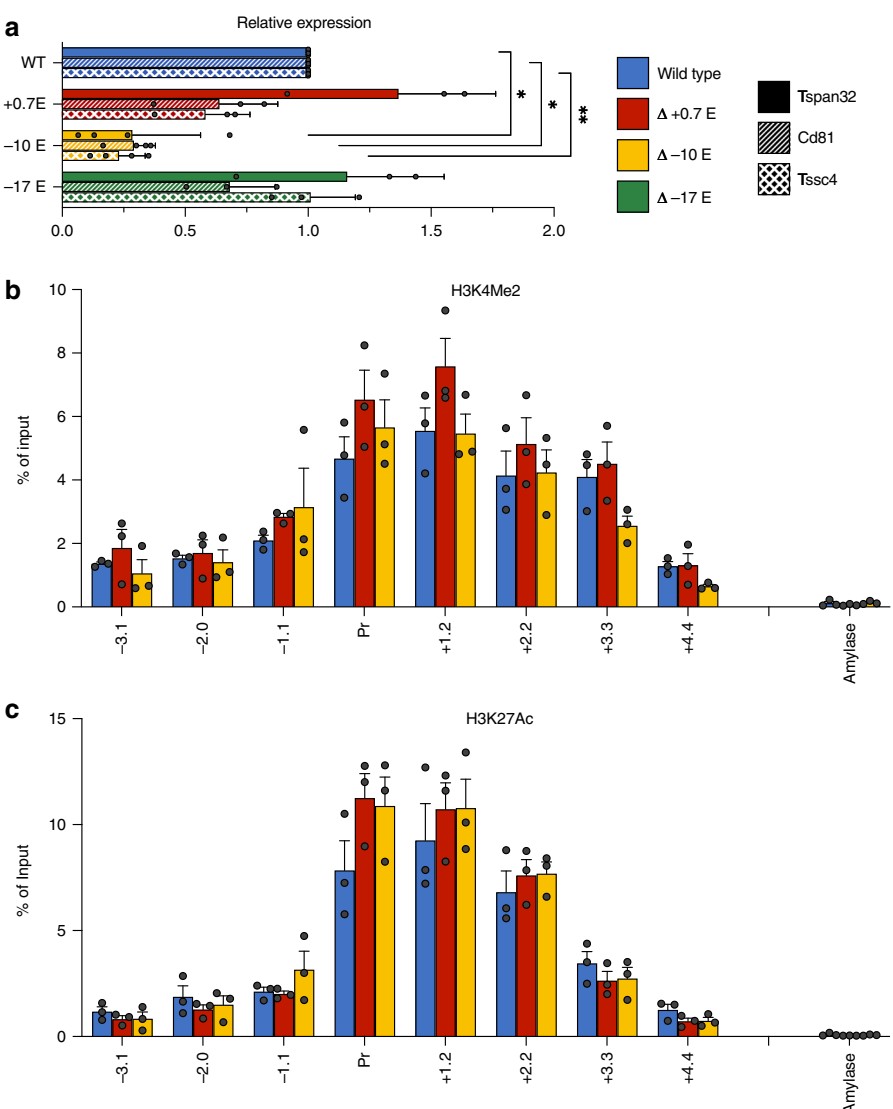

**Fig. 6 Effects of deletions of putative enhancers in the *Tspan32* gene locus in proliferating MEL cells. a** Normalized gene expression for *Tspan32, Cd81* and *Tssc4* measured by qrt-PCR in wt (MEL) and upon deletion of the indicated enhancer regions (0.7, −10, −17). *P*-values were calculated using a two-tailed Student's *t*-test; *$P < 0.05$, **$P < 0.005$, ***$P < 0.0005$. **b**, **c** Bar graphs showing % of input control for the indicated probes derived from qrt-PCR analysis of ChIPs using antibodies specific for H3K4Me2 (**b**) or H3K27Ac (**c**). Amylase indicates a control probe within the inactive amylase gene locus. Results of all panels are means + s.e.m of at least three independent experiments. Each colored circle represents the average of three technical replicates of an independent single-cell-derived homozygous enhancer knockout clone.

using widely available and robust antibodies — serves to identify genes that are both more highly expressed and more closely aligned to cell type than genes identified by super-enhancer analysis. Moreover, the protocol for classifying regions by peak breadth and identifying HCDs, which we include here (Supplemental Fig. 1, Supplementary Software 1), is both more transparent and simpler to use than the ROSE algorithm.

Notably, peak breadth in histone modification ChIP-seqs has previously been used to identify highly expressed and/or cell type-specific genes. Stretch enhancers, for example, have been defined as regions exhibiting continuous epigenomic signatures (chromatin state from ChromHMM)[29] indicative of enhancers over spans of 3 kb or more[30]. This contrasts with our own analysis in two key respects. First, hyperacetylated domains encompass both enhancers and promoters and are not necessarily limited to either type of sequence element. Second, the stretch enhancer definition applies to fully 10% of all putative enhancer regions, while our own definition of a hyperacetylated domain is more restrictive.

There are several potential explanations for the differences in genes and pathways identified by HCD (peak breadth) vs super-enhancer analysis. Assignment of genes to super-enhancers relies on the assumption that a given enhancer regulates the nearest active gene. This is, at best, an approximation, the accuracy of which is impossible to evaluate in the absence of genetic analysis or other data, and difficult even then. In contrast, genes assigned to HCDs are nearly always (95%) located within them, and there is a logically greater certainty that a gene within an HCD is directly affected. Complexity arises in cases involving multiple active genes within an HCD, and so even the domain concept involves a degree of approximation, but based on gene expression and cell type-specificity (Figs. 2, 3), this uncertainty appears to be less than that inherent to super-enhancers.

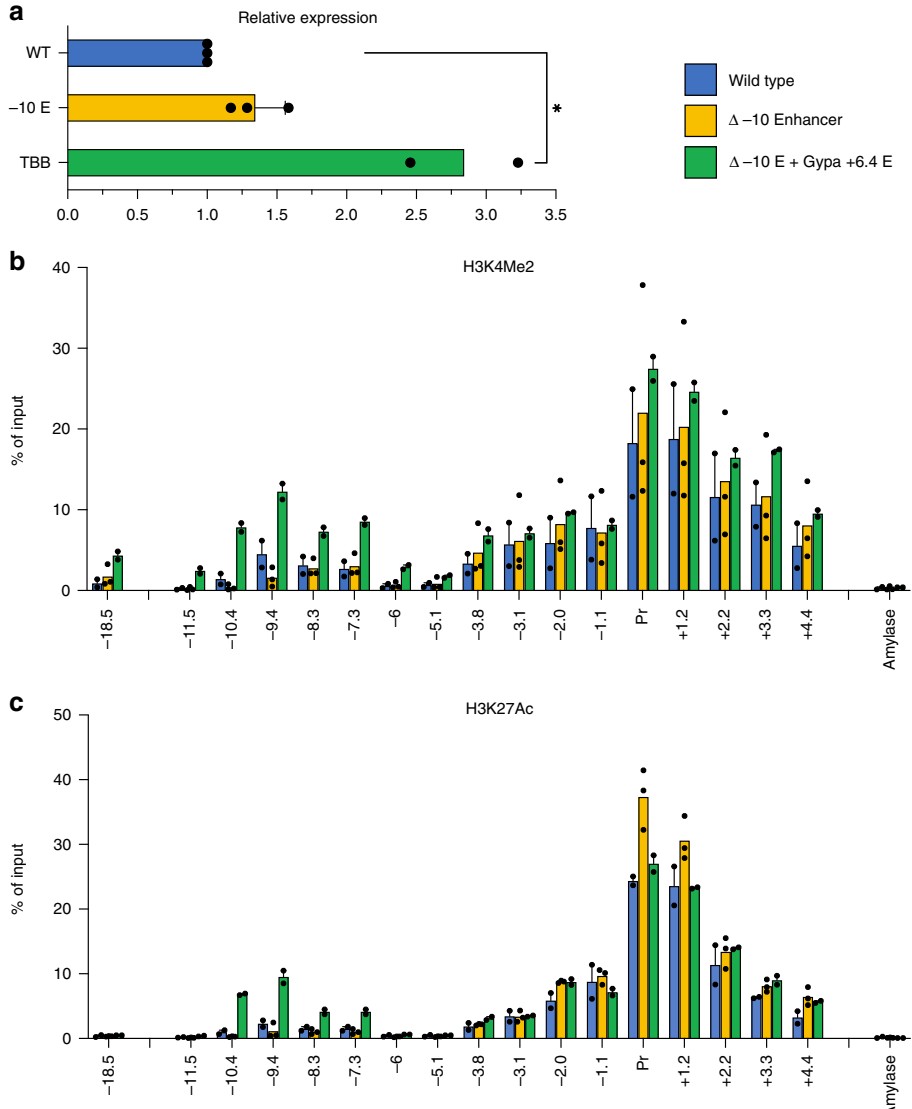

**Fig. 7 Effects of addition of the *Gypa* + 6.4 enhancer in the *Tspan32* gene locus in differentiating MEL cells. a** Normalized gene expression for *Tspan32* measured by qrt-PCR in wt (MEL) and upon deletion of the −10.0 enhancer or insertion of the *Gypa* + 6.4 enhancer. *P*-values were calculated using a two-tailed Student's *t*-test; *$P < 0.05$. **b**, **c** Bar graphs showing % of input control for the indicated probes derived from qrt-PCR analysis of ChIPs using antibodies specific for H3K4Me2 (**b**) or H3K27Ac (**c**). Amylase indicates a control probe within the inactive amylase gene locus. Results of all panels are means + s.e.m of at least three independent experiments for WT and Δ −10.0 Enh and two independent experiments for Δ −10 E + Gypa +6.4 E. Each colored circle represents the average of three technical replicates of an independent single-cell-derived homozygous enhancer knockout clone.

To a certain extent, however, HCDs may be advantaged compared to super-enhancers due to technical aspects of ChIP-seq analysis. Ranking by peak intensity (e.g., peak height) is limited because mapping algorithms routinely discard duplicate reads as a precaution against PCR artifacts arising from library construction. Reads therefore cannot stack on top of each other, and the maximum height of a given peak is artificially limited. Ranking by peak breadth avoids this limitation and therefore may present a more sensitive metric for determining the strongest and/or most cell type-specific enhancers.

An important consideration is the potential mechanisms that underlie differences in peak intensity vs differences in peak breadth. In ChIP-seqs, which assay populations of cells, differences in peak intensity most likely reflect differences in binding affinity between enhancer regions; thus, higher peak intensity for a TF reflects a higher proportion of alleles crosslinked to that

factor, and for histone modifications a higher proportion of alleles associated with the cognate enzyme(s). In either case, binding does not necessarily invoke transcriptional activation of a neighboring gene.

In contrast, the largest peak breadths most likely arise from a mechanism distinct from higher binding affinity. We previously demonstrated that within the murine β-globin locus association of specific histone modifications was not intrinsic to the entire sequence underlying the HCD, but was a function of a smaller enhancer region within it. We demonstrate similar effects for enhancers within the *Gypa* and *SLC4A1* gene loci in this study. This requirement suggests that the histone modification pattern associated with an HCD spreads in some fashion from the regulatory element(s). Regardless of the mechanism that underlies such spreading — whether a specific mechanism akin to heterochromatic domain formation, or simple spill-over from an

excess of chromatin modifiers — a broad domain implies a high degree of binding and activity, and insofar as the vast majority of HCDs subsume gene promoters, of activity at transcription initiation sites. Rather than implying that a greater proportion of alleles in the cell population harbor modified nucleosomes, an HCD implies a greater quantity of modified histones at each allele, which could contribute to stability of the active transcriptional state in the nuclear environment.

A recent study examined the specificity of antibodies directed against different methylated states of H3K4, and determined that many of them exhibited cross-reactivity[31]. Among the antibodies we have used, this study indicated that the H3K4Me1 Ab was highly specific, while the H3K4Me2 Ab exhibited significant cross-reactivity with H3K4Me1 and the H3K4Me3 Ab with H3K4Me2. All of these reagents were polyclonal antisera, however, and so differences in the behavior of different lots of these antibodies introduce an additional level of uncertainty. We are therefore cautious in interpreting the biological significance of the H3K4Me2 enrichments we show here, outside of the apparent role of specific enhancers in contributing to them. Notably, of the other ChIP-seq data sets we have analyzed, one used the same H3K4Me2 Ab as here (see Methods), while the other two used a distinct reagent. Based on a search of available databases, these two reagents account for more than 80% of H3K4Me2 ChIP-seqs. Our ability to consistently identify HCDs from ChIP-seqs using these antibodies suggests that differences in specificity alone do not account for the utility of the H3K4Me2 antibody, and the HCD approach produces similar results with other antibody combinations.

Notably, ranking of H3K27Ac/H3K4Me2 peaks by breadth is potentially useful in identifying enhancers that function by different mechanisms. In this study, we contrast the effects of the deletion of enhancers from within the endogenous *Gypa*, *SLC4A1*, and *Tspan32* gene loci, which are all classified as super-enhancers, but only the *Gypa* and *SLC4A1* loci as HCDs. At all loci, enhancer deletions result in substantial decreases in gene expression. At the *Gypa* and *SLC4A1* loci, deletion of the enhancers results in loss of the broad peak of histone hyperacetylation, including modification levels associated with the region proximal to the TSS. At the *Tspan32* locus, however, the deletion of neither the −10 kb element in undifferentiated MEL cells nor the +0.7 element in differentiating MEL cells results in any effect on chromatin structure at gene promoters or the regions surrounding them. Moreover, the introduction of the *Gypa* enhancer into the *Tspan32* locus, at the site of the deleted −10 kb element, results in the formation of an HCD over a region of the locus that normally does not exhibit one. The data suggest a functional distinction between enhancers that work, at least in part, via modulation of long-range chromatin structure and enhancers that do not.

The relationship between enhancer function and HCD formation, however, does not appear to follow the simplest possible model. Both of the putative enhancers within the *SLC4A1* locus are required for HCD formation, despite a separation of nearly 12 kb between them. This collaborative function raises the question of whether HCD formation by the *Gypa* enhancer inserted within the *Tspan32* locus is fully enhancer-autonomous, or if instead it requires additional elements, both at this locus and at its native location. Additional genetic manipulations will be required to address this question.

An additional concern is the stitching step of the super-enhancer analysis, in which putative enhancer regions located within a specific distance of each other (the default in the ROSE algorithm is 12.5 kb) are considered as a single element, under the assumption that closely spaced enhancers work together in activation of the same gene or genes[6–8]. Some studies have questioned the general validity of this assumption[10,32], and our own

analysis of the *Tspan32* locus presents a clear example of how the super-enhancer concept tends to oversimplify more complex patterns of gene regulation. We found that the *Tspan32* super-enhancer, comprised of three distinct candidate enhancer regions, is in fact involved in the regulation of two genes in addition to *Tspan32*. Moreover, individual enhancers within the super-enhancer exhibit different roles, with the −10 kb enhancer required solely in undifferentiated MEL cells, the +0.7 kb enhancer required solely in differentiating MEL cells, and the −17 kb enhancer showing no requirement at all. Based on such behavior, the classification of these three elements as a single super-enhancer appears to obscure the functions of these regions more than it illuminates them.

While super-enhancers can identify highly expressed genes that characterize cell identity, we find that HCDs define a set of genes that exhibit higher expression, and better specify cell lineage. Furthermore, as an analytical tool, HCDs present additional advantages over super-enhancers, including a significantly simpler workflow. Our results suggest that super-enhancers may not have a function distinct from that of typical enhancers, and that clustering of enhancer elements does not necessarily imply cooperative function, whereas there appears to be a functional difference between at least some of the enhancers located within HCDs, involving modulation of long-range chromatin structure, compared to other enhancers.

## Methods

**Mice and tissues.** Mice (C57BL/6J) were mated overnight and the vaginal plug verified the next morning, indicating embryonic day 0.5 (e0.5). At e14.5 pregnant mice were killed by cervical dislocation for collection of fetal liver. The Institutional Animal Care and Use Committee affiliated UCAR at the University of Rochester Medical Center has reviewed and approved all protocols involved in this project for the use of mice. (UCAR 2006-119). For ATAC-seq (see below), E14.5 liver-derived proerythroblasts were sorted as previously described[33], but also including Cd117 staining using a FACS Aria-II. In brief, larger single cells that were Ter$^{lo}$ CD117+CD44$^{hi}$ were collected. These cells are also positive for CD71.

**ChIP-seq.** Chromatin immunoprecipitation was performed as previously described[34]. In brief, $2 \times 10^7$ cells were washed with PBS once, then cross-linked with 1% formaldehyde for 10 min at room temperature. Cross-linking was quenched with 5 M glycine for 1 min at room temperature, and cells were washed with PBS once. Cells were incubated in swelling buffer for 20 min on ice, followed by dounce homogenization to isolate cross-linked nuclei. Nuclei were placed in lysis buffer for 30 min and then sonicated into ~200 bp fragments using a Diagenode Bioruptor. Samples were diluted and immunoprecipitated with 200 μg antibody to either H3K4Me1 (Abcam #ab8895), H3K4Me3 (Active Motif #39916), H3K27Ac (Active Motif #39134), H3K4Me2 (Abcam #7766), or nonspecific rabbit immunoglobulin G (Millipore #12-370) and incubated on a rotator for 18 h. at 4 °C. DNA-protein complexes were recovered with protein G magnetic beads (Invitrogen). Library preparation was performed as previously described[35]. Library quality was evaluated on a Bioanalyzer and sequencing was performed on a HiSeq 2500 Rapid Run to obtain $1 \times 50$ bp reads.

**ChIP-Seq analysis.** Illumina reads were converted to the fastq format using bcl2fastq-1.8.4 with default parameters. Quality control and adapter removal was performed using Trimmomatic-0.32 (ref. [36]) using the following parameters SLI-DINGWINDOW:4:20 TRAILING:13 LEADING:13 ILLUMINACLIP:adapters. fasta:2:30:10 MINLEN:15. All quality reads were aligned to the mm9 reference genome using Bowtie-1.0.1 (ref. [37]), suppressing multi-mapping reads using the '-m 1' parameters. All alignments were written in the SAM format (-S), converted to BAM and sorted for all subsequent analyses using samtools[38]. Peaks were called for each replicate of each mark using MACS2 along with additional settings including (–broad –broad-cutoff 0.1 -B) using the total input control as the mock data file[16]. The intersection of each replicate was identified using bedtools intersect and was used for subsequent analyses[39].

**ATAC-seq.** ATAC-seq was performed as previously described[40]. In brief, $5 \times 10^4$ sorted proerythroblasts were lysed by gently pipetting in cold lysis buffer. Cell lysate was resuspended in a transposition reaction mix (Illumina) and incubated at 37 °C for 30 min. Reactions were purified using AmpureXP beads (Beckman Coulter) following the manufacturer's protocol with minor changes. Beads were used at a 1:1.1 ratio and reactions were washed twice. After purification samples were amplified using $1 \times$ NEBnext PCR master mix and 1.25 μM of custom Nextera

PCR primers 1 and 2 (ref. [41]). Libraries were amplified again for an additional 17–19 cycles and left side size selected with SPRIselect beads (Beckman Coulter) at a 1:1 ratio following manufacturer's protocol, then right side size selected with SPRIselect beads (Beckman Coulter) at a 1:0.5 ratio following manufacturer's protocol. Library quality was evaluated on a Bioanalyzer and sequencing was performed on a Hiseq2500v2 platform in rapid mode to generate 50 million reads per sample.

**ATAC-seq analysis**. Illumina reads were converted to the fastq format using bcl2fastq-1.8.4 with default parameters. Quality control and adapter removal was performed using Trimmomatic-0.32 (ref. [36]) using the following parameters SLI-DINGWINDOW:4:20 TRAILING:13 LEADING:13 ILLUMINACLIP:adapters. fasta:2:30:10 MINLEN:15. All quality reads were aligned to the mm9 reference genome using Bowtie-1.0.142 (ref. [37]), suppressing multi-mapping reads using the '-m 1' parameters. All alignments were written in the SAM format (-S), converted to BAM and sorted for all subsequent analyses using samtools[38]. Reads aligning to organism-specific blacklist regions and the mitochondrial genome are discarded. Accessible regions are identified using MACS2 (ref. [15]) and ATAC specific parameters (–nomodel –shift -100 –extsize 200).

**RNA-Seq analysis**. Raw fastq files for publicly available data sets (GSE87064-SRR4253101, SRR4253102, GSE98724- SRR5520174 SRR55201745) were downloaded using fastqDump available in SRAtoolkit[21–23]. All reads were processed using trimmomatic[36] (v0.36) to remove low quality bases and any residual adapter sequence (TRAILING:13 LEADING:13 ILLUMINACLIPtrimmomatic_adapters. fasta:2:30:10 SLIDINGWINDOW:4:20 MINLEN:15). Quality reads were aligned to either mm9/hg19 (depending on organism of origin) using STAR[42] (v2.5.2b) (STAR –twopassMode Basic –readFilesCommand zcat –runThreadN –runMode alignReads –genomeDir –readFilesIn –outSAMtype BAM Unsorted –outSAMstrandField intronMotif –outFileNamePrefix –outTmpDir –outFilterIntronMotifs RemoveNoncanonical –outReadsUnmapped Fastx). Read alignments were quantified using featureCounts (v1.5.0-p3) (-s 2 -T -t exon -g gene_name -a genes.gtf –o) and normalization was performed within the R (3.4.1) framework using DeSeq2 (v1.16.1)[43].

**HCD analysis**. The top 2% broadest peaks were filtered from the H3K4Me2 and H3K27Ac intersected replicates. Bedtools intersect was used to identify the intersection of these two tracks, which is what we define as a hyperacetylated domain[37]. The code to call HCDs is included in Supplementary Software 1. HCDs were associated with nearby genes using bedtools closest, default settings. Microarray expression data for e14.5 fetal liver (maturational stages proerythroblast, basophilic erythroblast, and polychromatic erythroblast) available through ErythronDB were downloaded and used to associate nearby gene expression to domains and non-domains[16,17]. Enrichr was used to perform comprehensive enrichment analysis on associated genes[18,19].

**Super-enhancer analysis**. For the identification of super-enhancers within our H3K4Me2 and H3K27Ac data sets we used ROSE (http://younglab.wi.mit.edu/super_enhancer_code.html), created by the Young lab[6]. Input enhancers were defined as a merged peak set of all Gata1 and Tal1 replicates[44] and a TSS_EX-CLUSION_ZONE_SIZE of 500 bp. Super-enhancers were associated with nearby genes using bedtools closest, default settings. Enrichr was used to perform comprehensive enrichment analysis on associated genes[18,19].

**Me3 domain analysis**. The top 5% broadest peaks were filtered from the H3K4Me3 intersected replicates. Domains were associated with nearby genes using bedtools closest, default settings. Enrichr was used to perform comprehensive enrichment analysis on associated genes[18,19].

**Analysis of human ChIP-Seq and microarray data**. Domains and super-enhancers were identified and evaluated during human fetal erythropoiesis using publicly available data (GSE36985)[20]. Mapped read bed files were downloaded and converted to the BAM format (bedtobam bedtools) for fetal H3K4Me2, H3K27Ac, Gata1, Tal1 and the total input control samples. MACS2 was used to identify histone marks (-B –broad –broad-cutoff 0.1) and TF peaks (-B –q 0.01) relative to the total input control[15]. Domains were defined as the intersect of the largest 2% H3K4Me2 (regions larger than 8,015 bp, 936 regions) and H3K27ac (regions larger than 9,020 bp, 503 regions), for a total of 302 regions. Two super-enhancer populations were called using ROSE and identified based on a Gata1/Tal1 union enhancer population which was ranked on H3K4Me2 or H3K27Ac[6]. The resulting sets of super-enhancers were intersected to identify a final population of super-enhancers (326). Domains and super-enhancers were associated with hg18 RefSeq Genes (UCSC) based on a nearest neighbor analysis using bedtools closest (default settings)[33]. Enriched cell types were evaluated using Enrichr and the Human Gene Atlas. RNA expression log2(RMA) were downloaded from ArrayExpress (G_GEOD-36984)[18–20].

**CRISPR deletion and insertion**. px458 plasmids harboring sgRNAs targeting specific regions of the mouse genome were engineered as previously described[45]. In brief, sgRNAs were designed following guidelines to minimize off-target effects (http://crispr.mit.edu/) (Supplementary Table 1). Oligonucleotides were annealed as follows: 10 μM guide oligo, 10 μM reverse complement guide oligo, 1x T4 ligation buffer, 5U T4 polynucleotide kinase (New England Biolabs) were heated to 37 °C for 30 min, 95 °C for 5 min, and cooled to 25 °C at a rate of 5 °C min$^{-1}$. The annealed oligos were cloned into pSpCas9(BB)-2A-GFP(px458) (Addgene ID 48138) using 100 ng px458 cut with BbsI (New England Biolabs), 0.2 μM annealed oligos, 1× quick ligase reaction buffer (New England Biolabs), and 2000U quick ligase and incubated at room temperature for 10 min. For deletions, MEL745a cells were transfected with one plasmid targeting upstream and one plasmid targeting downstream sequence surrounding the enhancer using Lipofectamine 3000 (Invitrogen) according to manufacturer instructions. For insertions, MEL745a cells were transfected with one plasmid targeting the sequence surrounding the previously deleted enhancer and a double-stranded repair template using Lipofectamine 3000 (Invitrogen) according to manufacturer instructions. At 48 h. the cells were resuspended in D-PBS (Gibco) +0.5% FBS(Gemini), stained with DAPI (ThermoFisher), and the viable, GFP positive population was sorted in bulk (Supplementary Fig. 11). Seven days after sorting, cells were diluted to 1 cell/100 μl in Dulbecco modified Eagle medium (Gibco) containing 20% FBS, 1% Glutamax (Gibco), and 1% Pen/strep (Gibco) and plated in a 96-well plate. Homozygous deletions were identified through PCR of the region surrounding the targeted enhancer sequence, and PCR products were Sanger sequenced for confirmation.

**Cell culture**. MEL745a cells were obtained from the laboratory of Mark Groudine. Cell lines were maintained at 37 °C in a $CO_2$-humidified atmosphere. MEL745a cells were cultured in Dulbecco modified Eagle medium containing 10% FBS, 1% Glutamax, and 1% Pen/strep. For induction, 2% dimethyl sulfoxide (Fisher) was added to the culture medium.

**ChIP-qPCR**. Cells were formaldehyde crosslinked, sonicated and immunoprecipitated, and DNA isolated as for the ChIP-sequencing, with minor differences. Nuclei were isolated from $1 \times 10^7$ cells for each experiment and then sonicated to ~500 bp fragments of genomic DNA using a Diagenode Bioruptor. Samples were diluted and immunoprecipitated with 5 μg of antibodies specific for H3K27Ac (Active Motif # 39134), H3K4Me2 (Millipore # 07-030), or nonspecific rabbit immunoglobulin G (Milipore # 12-370). The analysis was performed using qPCR and detected using the CFX Connect Real-Time PCR System and CFX Manager (Bio-Rad). Primers were designed to amplify regions within the Glycophorin A locus or *Tspan32* locus, and as a negative control the *Amylase* gene promoter (a region that is inactive in erythroid cells) (Supplementary Table 2).

**qRT-PCR**. RNA was isolated using TRIzol (Invitrogen) according to manufacturer instructions. One microgram of RNA was used to synthesize cDNA using the iScript cDNA synthesis kit (Bio-Rad). cDNA was amplified using iTaq Universal SYBR Green Supermix (Bio-Rad) and detected using the CFX Connect Real-Time PCR System and CFX Manager (Bio-Rad). Primers were designed to amplify *Gypa*, *Tspan32*, *Cd81*, and *Tssc4* cDNAs, and also ribosomal 18S as a control (Supplementary Table 3).

**Statistics and reproducibility**. Gene expression and ChIP-qPCR experiments were performed on three biological replicates with the exception of the *Gypa* promoter deletion as well as the enhancer replacement experiment where only two biological replicates of the homozygous insertion of the new enhancer were found (standard error of the mean (S.E.M.) is shown). Individual data points represent the mean of three technical replicates. Biological replicates are defined as experiments performed on an independent single-cell-derived homozygous knockout clone. Technical replicates are defined as experiments performed on the same sample and analyzed multiple times. Statistical significance for gene expression was calculated using a two-tailed Student's *t*-test.

**Reporting summary**. Further information on research design is available in the Nature Research Reporting Summary linked to this article.

## Data availability

All ChIP-seq and ATAC-seq data generated in this study are available through GEO: GSE132130. Additional ChIP-seq, ATAC-seq, and RNA data sets were downloaded from GEO[20–23,44]. Mouse erythroid microarray data were downloaded from ErythronDB[16,17]. Human microarray data were downloaded from ArrayExpression[20]. The source data underlying the remaining data are provided as a source data file. All data are available from the corresponding author upon reasonable request. Source data are provided with this paper.

## Code availability

Code to call HCDs is included in Supplementary Software 1. Source data are provided with this paper.

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

## Acknowledgements

The authors thank Laurie Steiner and Patrick Murphy for critical reading of the manuscript. This work was supported by NIH R01DK070687. S.F. was supported by Grant Number GM068411 from Institutional Ruth L. Kirschstein National Research Service Award.

## Author contributions

M.B. initiated, conceived, designed, and supervised the research, analyzed data and wrote the paper. S.F. designed and performed most experiments, analyzed data, generated figures, and edited and revised the paper. J.A.M. designed experiments, analyzed data, generated figures and edited the paper. C.D., M.G., P.D.K., and N.F. performed experiments.

## Competing interests

The authors declare no competing interests.
