## [Peer Review File · Nature Communications]

Reviewers' comments:

Reviewer #1 (Remarks to the Author):

In this paper, Fox et al use H3K27ac breadth and H3K4me2 enrichment to identify key enhancers in erythrocytes. They call these regions hyperacetylated chromatin domains (HCDs). They go on to show that HCDs are associated with high levels of gene expression (nearest neighbour analysis) that are also more tissue specific than genes called using superenhancers (SEs). Finally, to try and functionally distinguish SEs from HCDs, they delete one example of an HCD and one example of a SE and show that although both have effects on gene regulation, only the HCD deletion impacts H3K27ac.

I appreciate the attempt to functionally distinguish HCDs from SEs, but in its present form the work seems to be too preliminary to draw robust conclusions. More specifically, I have a few major and several minor concerns about the study:

Major points

1. It is not clear to me what the underlying functional importance of HCDs may be. Is H3K27ac breadth a cause or consequence of enhancer activity? Is H3K27ac non-functional at SEs compared to HCDs?
2. The comparison of the Gypa HCD and the Tspan32 SE is interesting, but with only one of each class to compare, it is difficult to know how far one can generalize about SE's having little effect on chromatin structure while HCDs function mainly through altering histone modifications such as H3K27ac. Is the HCD at Gypa a special case or do others function in the same way? Are all SEs similarly inert when it comes to changes in H3K27ac? Also, this relates to point 1 above, the observation of HCD dependent changes in H3K27ac are descriptive and it is not clear what the underlying mechanism might be.
3. Do better characterized SEs behave the same way as Tspan2? For example, MYC has a well known SE 2Mb away from the gene that requires BRG1 binding and H3K27ac to maintain its function (Shi et al Genes Dev. 2013 27(24):2648-62. doi: 10.1101/gad.232710.113 and Bahr et al Nature 2018 doi:10.1038/nature25193). How do deletions of this enhancer impact H3K27ac levels and how does this impact the underlying mechanism of how this enhancer is thought to function? What about genes like Mitoferrin and a-globin that "look" like SEs, but may not be?
4. To identify important tissue specific genes, H3K4me3 peak breadth (Benayoun et al Cell. 2014;158(3):673-88) was shown to be superior to SEs for identifying genes that were important for cell identity. One of the issues with connecting HCDs to genes is that enhancers sometimes regulate more than one gene. Calling broad H3K4me3 peaks has a distinct advantage in that it automatically links up genes to peaks, as this is mainly a promoter based approach. In addition, it has the advantage of requiring only a single histone mark for analysis. Are HCDs superior to calling broad H3K4me3 peaks as a way of identifying key tissue specific gene targets?

Minor points

5. H3K27ac is a mark of promoters as well as enhancers. The examples of HCDs shown in Figure 1 appear to be continuous with the promoter regions. Are all HCDs distinct enhancers or are they just strong promoters with H3K27ac "spillover"?
6. Since only PCR is use to quantify the ChIP in Figure 4 and 5 it is hard to conclude that there are no

changes in H3K27ac in the SE deletion, as they could simply be looking at the wrong region.

7. SEs are also useful for identifying key oncogenes in cancer cells. Are HCDs better or similar to SEs for identifying enhancers that drive oncogenesis?

8. The identification of SEs now relies more on MED1 and BRD4 enrichment, as well as H3K27ac. Would using these parameters identify a similar set of enhancers as HCDs?

9. It may seem slightly wasteful to do genome wide experiments on the enhancer deletion mutants, but considering the long range effects that the Tspan32 enhancer deletion has on multiple genes in the region, it would be interesting to have ChIP-seq and RNA-seq for the deletion mutants to assess regulation of multiple genes in the region.

10. The H3K4me2 broad domains in Figure 1b-c are quite striking. Other H3K4me2 ChIP-seq tracks I have seen don't display such robust enrichment at active genes. What evidence is there that the antibody used in these experiments is truly specific to H3K4me2?

11. Rather than just the metanalysis shown in Figure S2 and S3, it would be useful to have examples tracks of HCDs from published datasets in other tissues. Also, there must be many more H3K4me2 and H3K27ac datasets out there, it would be useful to see how well calling HCDs performs in multiple human tissues for identifying key tissue specific target genes.

12. It would be useful to see the ChIP-seq tracks and the position of the PCR primers on the tracks for the enhancers studied in Figures 4-6 to get a better sense of how well these ChIP PCR experiments cover the locus.

Reviewer #2 (Remarks to the Author):

The manuscript "Hyperacetylated Chromatin Domains Mark Cell Type-Specific Genes and Suggest Distinct Modes of Enhancer Function" by Fox and colleagues describes characterization and validation of a novel strategy to classify regions of hyperacetylated chromatin domains associated with histone marks of active genes. This technique performed better than algorithms to identify "super enhancers" and was able to identify genes that are both more highly expressed and more relevant to cell identity and function.

General Comments:

Regions of acetylated histones have been known to be associated with active gene expression for some time, with hyperacetylated regions, as described by several groups including this one and the one in Seattle, containing genes critical for cell state and function. The Broad group labeled these regions as super-enhancers with a 12.5kb window definition in the Rose algorithm. Another name was given by the Cal and NIH groups, stretch enhancers, refining the definition based on further characterization of chromatin state. The technique developed in this report performs better than the Rose algorithm used to identify "super enhancers." In addition, it was able to identify genes that are both more highly expressed and more relevant to cell identity and function than Rose and related tools. This technique refines identification of active enhancers using only acetylation of lysine 27 of histone 3 and demethylation of lysine 4 of histone 3 without the numerous histone marks required by the ChromHMM stretch enhancer. Thus the technique is of wide interest and applicability to many (?all!) cell types and is a major advance in our identification and definition of hyperacetylated chromatin domains, with improvements in relevant functional correlates.

The manuscript is well written and the data support the hypotheses. The figures and Supplemental data are clearly presented and contribute to the report. The validation data are straightforward. They support the clear observation that enhancers, whatever they are called, are complex, and when located in HCD, exhibit high degrees of complexity. This is an important point of this report, perhaps downplayed to avoid the "super" versus "not so super" enhancer controversy.

Specific Comments:

1. Where do transcription factors fit into this equation? Are TF enriched, especially erythroid lineage specific TF? Are the authors able to differentiate the non-promoter (enhancer) sites from the promoter sites to identify whether specific motifs associate with subsets of induced genes at HCDs? It may allow identification of other (novel?) regulators besides the known GATA1, KLF1 etc. active in erythroid cells.
2. The authors note their data demonstrate the ever emerging complexity of enhancers. Did any of their analyses provide insight into further characterization of these active, hyperacetylated domains? Were they near poised gene promoters? Did cis-element motifs have location preference, such as at the center of H3K27ac or at the edges of H3K4me3, H3K9me3, etc., similar to Epigram.
3. Comparison to Rose. Attempting to seek some granularity, it is logical to assume these HCD were at sites of Rose-identified SE. Many of the top regions identified by Rose are at sites critical for cell identity, containing genes encoding important cell identity determinants, signaling molecules, TF, etc. How did the author's technique compare, both at the specific region level, e.g. around GATA1 or KLF1 gene loci for example and at a broad scale?
4. Comparison to Stretch E. With the various histone marks the authors downloaded (and are there more?), did the authors attempt to construct a ChromHMM-like dataset for comparison to their technique? It would be helpful to compare the two techniques.
5. How cell type-specific were HCD?
6. Were HCD aligned with other structural elements, CTCF, known TADs, etc. ?

Reviewers' comments:

Reviewer #1 (Remarks to the Author):

In this paper, Fox et al use H3K27ac breadth and H3K4me2 enrichment to identify key enhancers in erythrocytes. They call these regions hyperacetylated chromatin domains (HCDs). They go on to show that HCDs are associated with high levels of gene expression (nearest neighbour analysis) that are also more tissue specific than genes called using superenhancers (SEs). Finally, to try and functionally distinguish SEs from HCDs, they delete one example of an HCD and one example of a SE and show that although both have effects on gene regulation, only the HCD deletion impacts H3K27ac.

I appreciate the attempt to functionally distinguish HCDs from SEs, but in its present form the work seems to be too preliminary to draw robust conclusions. More specifically, I have a few major and several minor concerns about the study:

We appreciate the points that this reviewer has made, and have attempted to edit the manuscript, and add additional data, that we feel addresses the perception that the work is “preliminary” in nature (see below).

Major points

1. It is not clear to me what the underlying functional importance of HCDs may be. Is H3K27ac breadth a cause or consequence of enhancer activity? Is H3K27ac non-functional at SEs compared to HCDs?

We apologize for the confusion. “HCDs” are “functionally important” in the same way that super-enhancers are: they represent a stratification of ChIP-seq peaks that serves to identify highly-expressed, cell type-specific genes. As we note in the introductory paragraphs, however, the “super-enhancer” strategy has not in addition served to identify functional differences between enhancers that are “super” and other enhancers. In contrast, we now have a handful of examples of enhancers located within HCDs that function, at least in part, by mediating the formation of an HCD. Other enhancers do not appear to do this, and so the behavior of the enhancers found within HCDs that we have tested delineates a functional class of enhancer.

We have edited the text to make clearer the point that the broad peak of enrichment for selected histone modifications is a consequence of enhancer activity. As for H3K27ac, we do not generalize in any way regarding the importance of this modification for SEs; the GYPA enhancer, for example, is also an SE (see Fig. 1). In fact, we do not make generalized conclusions about SEs at all (see also below). Nevertheless, however the enhancers at the Tspan32 locus are working, it apparently does not involve regulation of H3K27ac levels

surrounding the gene promoter, and so this modification at the promoter alone – while potentially still important for gene activation – is not sufficient. We have attempted to clarify these points in the text.

2. The comparison of the Gypa HCD and the Tspan32 SE is interesting, but with only one of each class to compare, it is difficult to know how far one can generalize about SE's having little effect on chromatin structure while HCDs function mainly through altering histone modifications such as H3K27ac. Is the HCD at Gypa a special case or do others function in the same way? Are all SEs similarly inert when it comes to changes in H3K27ac? Also, this relates to point 1 above, the observation of HCD dependent changes in H3K27ac are descriptive and it is not clear what the underlying mechanism might be.

To address this criticism we now provide an additional example of an HCD, at the SLC4A1 (Band 3) gene locus, within which separate deletions of 2 enhancers demonstrate effects on long-range histone modifications. We unfortunately do not have the resources to design and implement a high-throughput strategy for deletions of enhancers within HCDs and/or other enhancers to generalize the principle further. The conclusion, however, is not that SEs in general fail to have any effect on chromatin structure – in fact, there is significant overlap between the two categories, and so some SEs are also HCDs. The main conclusion is that thus far all of the HCD-associated enhancers we have tested – coupled with a handful of additional published examples^{1,2} – show effects on long-range chromatin structure, while the Tspan32 SE elements provide an example of a different class of enhancer that has no effect on long-range chromatin structure. Thus we are not trying to make any sweeping generalizations about super-enhancers. We have attempted to clarify this in the text.

1. Fromm, G. *et al.* An embryonic stage-specific enhancer within the murine b-globin locus mediates domain-wide histone hyperacetylation. *Blood* **117**, 5207–5214 (2011).
2. Ho, Y., Elefant, F., Cooke, N. & Liebhaber, S. A defined Locus Control Region Determinant Links Chromatin Domain Acetylation with Long-Range Gene Activation. *Mol. Cell* **9**, 291–302 (2002).

3. Do better characterized SEs behave the same way as Tspan2? For example, MYC has a well known SE 2Mb away from the gene that requires BRG1 binding and H3K27ac to maintain its function (Shi et al Genes Dev. 2013 27(24):2648-62. doi: 10.1101/gad.232710.113 and Bahr et al Nature 2018 doi:10.1038/nature25193). How do deletions of this enhancer impact H3K27ac levels and how does this impact the underlying mechanism of how this enhancer is thought to function? What about genes like Mitoferrin and a-globin that “look” like SEs, but may not be?

Unfortunately, while both of the papers mentioned do show H3K27ac ChIP-seq data for the Myc-containing TAD and for the “blood enhancer cluster” (BENC), there is no corresponding data for the deletion of BENC that would indicate if Myc promoter-proximal

acetylation levels were affected. We are similarly unable to find relevant CHIP-seqs for deletions performed in the SLC25A37 (Mitoferrin) gene locus, or even at the alpha-globin locus for the systematic dissection of the putative SE there. Also, notably, none of these studies examine H3K4Me2 at all. Based on examination of the data shown in these publications, we would predict that these deletions have minimal effects on H3K27ac at the promoter regions, but there is no way to evaluate this prediction using the published data. Again, however, we are not attempting to draw any conclusions about the behavior of super-enhancers; we are merely showing examples of enhancers that do or do not affect long-range chromatin structure .

4. To identify important tissue specific genes, H3K4me3 peak breadth (Benayoun et al Cell. 2014;158(3):673-88) was shown to be superior to SEs for identifying genes that were important for cell identity. One of the issues with connecting HCDs to genes is that enhancers sometimes regulate more than one gene. Calling broad H3K4me3 peaks has a distinct advantage in that it automatically links up genes to peaks, as this is mainly a promoter based approach. In addition, it has the advantage of requiring only a single histone mark for analysis. Are HCDs superior to calling broad H3K4me3 peaks as a way of identifying key tissue specific gene targets?

To address these questions, we now include a comparison of the HCD strategy with H3K4Me3 peak breadth analysis, as performed in Benayoun et al. but using our own datasets. We find that HCDs only slightly outperform H3K4Me3 peak breadth in identifying highly-expressed genes. Meanwhile, H3K4Me3 peaks outperform HCDs in Enrichr combined score in erythroid tissues on the Mouse Gene Atlas. Although some terms are unique to HCDs, the H3K4Me3 method is arguably more effective at identifying genes associated with erythroid terms in functional enrichment analysis. This comparison is shown in Supplementary Figure 5. Importantly, however, the top 5% of H3K4Me3 peaks (the cutoff used by Benayoun et al.) subsumes fully 94% of HCD-associated genes. It therefore represents the HCD population and a large number of additional genes. Insofar as Enrichr combined score and functional enrichment analysis are cumulative, this explains the discrepancy. Advantages of HCDs over the H3K4Me3 method would therefore seem to be (1) a more parsimonious population of genes; and (2) the association of relevant enhancers, which are not necessarily encompassed by the smaller H3K4Me3 peaks.

As for identifying genes, as mentioned in the text, 95% of HCDs harbor active genes within them, and the remaining 5% harbor peaks of H3K4Me3 that suggest the presence of unannotated active promoters. In this respect, we would say that the accuracy of matching of genes to peaks is nearly equivalent between HCDs and the broadest H3K4Me3 peaks. There are often issues with multiple active genes within a single HCD (as we have noted in the text), but this is an issue with the broadest H3K4me3 peaks as well. Regardless, the near-total overlap of HCD-associated genes with the H3K4me3-associated genes would support the functional equivalence of the two approaches in this respect.

Minor points

5. H3K27ac is a mark of promoters as well as enhancers. The examples of HCDs shown in Figure 1 appear to be continuous with the promoter regions. Are all HCDs distinct enhancers or are they just strong promoters with H3K27ac “spillover”?

Based on the Gypa enhancer deletion, along with the SLC4A1 enhancer deletions we show in this revised manuscript, and previously published examples (notably HS-E1 within the β -globin locus), we would distinguish between H3K27ac that characterizes an HCD, and H3K27ac that is specifically directed to the promoter-proximal region. The loss of H3K27ac across the domains upon enhancer deletion strongly suggests that these enrichments are enhancer-dependent, whereas the promoter regions in each case do retain H3K27ac near the TSS (although the levels of this modification still decrease). We understand why this is confusing –First, H3K27Ac at a promoter within an HCD is engulfed within the HCDs domain of H3K27Ac, so that they are a single unit while the HCD is intact. Second H3K27ac is a feature of the mechanism that underlies enhancer-dependent HCD formation, but the data suggest that either the functional effect of the enhancer on the promoter does not involve this modification specifically, or that the “basal” (e.g. enhancer-deleted) level of this modification in the bulk population of alleles is not sufficient for high-level expression.

The reviewer may be asking this question in a different context, however (?), in that the suggestion is made that the HCD “enhancers” we characterize are actually just very strong promoters that also direct H3K27ac. The distinction between enhancers and promoters is the subject of current debate in the field, and this study is not meant to address this debate in any way. We would expect, based on current understanding, that the Gypa enhancer, for example, is the site of transcription initiation. It is, however, clearly an enhancer of the Gypa gene promoter, and one that directs H3K27ac over the gene body in a way that the gene promoter itself does not.

6. Since only PCR is use to quantify the ChIP in Figure 4 and 5 it is hard to conclude that there are no changes in H3K27ac in the SE deletion, as they could simply be looking at the wrong region.

The probes we use for qPCR analyses are 1-1.2 kb apart. Based on the resolution of the assay, we cannot exclude highly localized effects at the level of 1-3 nucleosomes. Nevertheless, it seems clear from the qPCR data that the Tspan32 enhancer deletions do not result in significant effects across the locus in general, as the enhancer deletion at the Gypa locus does. Regardless, any effect so highly localized would suggest a different mechanism from the one operative across the Gypa locus, and would thus only reinforce the main point.

7. SEs are also useful for identifying key oncogenes in cancer cells. Are HCDs better or similar to SEs for identifying enhancers that drive oncogenesis?

This is an excellent question, and one we have been interested in, but unfortunately do not have the resources to pursue fully. Any comparison of enhancers identified by HCD vs. SE will almost certainly define enhancers that are unique to HCDs and enhancers that are unique to SEs. The question of which enhancers are more significant in driving the transformed phenotype, however, is likely to be more uncertain than the assignment of enhancers to genes known to be relevant to specific primary cell types. We would need to perform experiments on novel pathways revealed by HCD analysis to evaluate this, and such a study would be well beyond the scope of the current manuscript.

8. The identification of SEs now relies more on MED1 and BRD4 enrichment, as well as H3K27ac. Would using these parameters identify a similar set of enhancers as HCDs?

Based on what we have encountered in the literature, MED1 would not identify HCDs. While we have not performed MED1 ChIP ourselves, we have not encountered broad peaks of association of this factor anywhere. Thus, a strategy based on ranking of peaks by breadth would not apply to a MED1 ChIP-seq. BRD4 has been shown to be associated with hyperacetylated histones generally, and so we would predict that peaks from BRD4 ChIP-seq could also be ranked by breadth. Unfortunately, the published BRD4 ChIP-seqs we have analyzed – mostly from erythroid cells – fail our QC tests and so we hesitate to perform any additional meta-analyses using them. That said, however, we question the assertion that current SE studies rely more on MED1 or BRD4; the majority of studies we encounter seem to utilize H3K27Ac. See, for example, Khan and Zhang, *Nucl. Acids Res.* 44:D164-71 (2016), in which the authors constructed a database of SE studies, in which 1 study (on mESCs) utilized MED1, 4 utilized TFs, and 95 utilized H3K27ac.

9. It may seem slightly wasteful to do genome wide experiments on the enhancer deletion mutants, but considering the long range effects that the Tspan32 enhancer deletion has on multiple genes in the region, it would be interesting to have ChIP-seq and RNA-seq for the deletion mutants to assess regulation of multiple genes in the region.

We unfortunately do not currently have the resources for this. As detailed in the text, however, we examined the TAD that contains Tspan32 and determined that it harbors 2 additional genes that are transcribed in erythroid cells; the effect of the deletions on these genes is shown in Figs. 5 and 6. We cannot rule out effects of the Tspan32 enhancers on genes outside of this TAD, but this would be an interesting finding in its own right, since the effect would cross TAD boundaries, and thus likely falls outside the scope of this study.

10. The H3K4me2 broad domains in Figure 1b-c are quite striking. Other H3K4me2

ChIP-seq tracks I have seen don't display such robust enrichment at active genes. What evidence is there that the antibody used in these experiments is truly specific to H3K4me2?

We have experience with this antibody for ~20 years. While we periodically test for specificity ourselves, this requires peptides or other resources that we currently do not possess, and so more recently we rely on the vendor (Active Motif, who perform an array of validation assays) or personal communications with other researchers for evaluation of specificity. The characteristics of qPCR-ChIP and ChIP-seq with this antibody, however, have remained consistent since we started using it, whether directly tested for specificity by us or not – that is, the broad peaks and robust enrichment we observe have not changed in any measurable way over that time span. We have qPCR-ChIPs of the *Gypa* and *SLC4A1* gene loci from 2004 (using murine fetal liver), with a fully vetted Ab, that are indistinguishable from the ones shown in this manuscript. Most of the publicly available ChIP-seq tracks we have seen are consistent with our own in this way as well. We have actually encountered H3K4Me2 tracks that exhibit mostly narrower peaks (as described by the reviewer) in published datasets, although not often. Barring cell-type differences, we would speculate that the differences reside more in the shearing and/or crosslinking steps of the ChIP procedure (in at least one case, narrower peaks characterized multiple antibodies targeting histone modifications in addition to H3K4Me2, suggesting a more general difference in the performance of ChIP). Regardless, we have no reason to believe that the H3K4Me2 ChIP-seqs we show in this study are any less specific than prior H3K4Me2 ChIPs we have performed.

11. Rather than just the metanalysis shown in Figure S2 and S3, it would be useful to have examples tracks of HCDs from published datasets in other tissues. Also, there must be many more H3K4me2 and H3K27ac datasets out there, it would be useful to see how well calling HCDs performs in multiple human tissues for identifying key tissue specific target genes.

This is an excellent suggestion and we would like to do this. There are two limitations to implementing this approach, however. First, the proportion of ChIP-seq datasets that include the H3K4Me2 mark, alongside the H3K27Ac mark, and an ATAC-seq or similar assay that allows for pinpointing of enhancer locations, is actually not that high. We included H3K4Me2 not because it is ubiquitously employed but because the antibody is so highly robust and consistent (as noted in #10, above), and therefore easily used in ChIP-seq assays. Second, a more comprehensive study of that sort would require more bioinformatician time and effort than we have at our disposal.

12. It would be useful to see the ChIP-seq tracks and the position of the PCR primers on the tracks for the enhancers studied in Figures 4-6 to get a better sense of how well these ChIP PCR experiments cover the locus.

This is an excellent suggestion, and the figures have been amended to show these features.

Reviewer #2 (Remarks to the Author):

The manuscript "Hyperacetylated Chromatin Domains Mark Cell Type-Specific Genes and Suggest Distinct Modes of Enhancer Function" by Fox and colleagues describes characterization and validation of a novel strategy to classify regions of hyperacetylated chromatin domains associated with histone marks of active genes. This technique performed better than algorithms to identify "super enhancers" and was able to identify genes that are both more highly expressed and more relevant to cell identity and function.

General Comments:

Regions of acetylated histones have been known to be associated with active gene expression for some time, with hyperacetylated regions, as described by several groups including this one and the one in Seattle, containing genes critical for cell state and function. The Broad group labeled these regions at super-enhancers with a 12.5kb window definition in the Rose algorithm. Another name was given by the Cal and NIH groups, stretch enhancers, refining the definition based on further characterization of chromatin state. The technique developed in this report performs better than the Rose algorithm used to identify "super enhancers." In addition, it was able to identify genes that are both more highly expressed and more relevant to cell identity and function than Rose and related tools. This technique refines identification of active enhancers using only acetylation of lysine 27 of histone 3 and demethylation of lysine 4 of histone 3 without the numerous histone marks required by the ChromHMM stretch enhancer. Thus the technique is of wide interest and applicability to many (?all!) cell types and is a major advance in our identification and definition of hyperacetylated chromatin domains, with improvements in relevant functional correlates.

The manuscript is well written and the data support the hypotheses. The figures and Supplemental data are clearly presented and contribute to the report. The validation data are straightforward. They support the clear observation that enhancers, whatever they are called, are complex, and when located in HCD, exhibit high degrees of complexity. This is an important point of this report, perhaps downplayed to avoid the "super" versus "not so super" enhancer controversy.

We thank the reviewer for this positive evaluation of the manuscript.

Specific Comments:

1. Where do transcription factors fit into this equation? Are TF enriched, especially erythroid lineage specific TF? Are the authors able to differentiate the non-promoter (enhancer) sites from the promoter sites to identify whether specific motifs associate with subsets of induced genes at HCDs? It may allow identification of other (novel?) regulators besides the known GATA1, KLF1 etc. active in erythroid cells.

We have attempted a number of analyses along these lines – by MEME-ChIP, etc. GATA1 is universal to (known) enhancers in erythroid cells no matter whether they are from SEs or HCDs or neither; we have looked for different arrangements of GATA1 binding sites to no avail. Other factors differ to some extent between different populations as we define them (e.g. enhancers associated with HCDs vs. with peaks of H3K27Ac/H3K4Me2 of 2 kb or less) but the significance associated with them is not very high. As the additional enhancer “replacement” experiment we include with the revised manuscript suggests, although there is clearly a domain-forming function that can be imported with the Gypa enhancer alone, the effect does not match what the enhancer does at its native locus, implying the contribution of additional elements within the locus. Such combinatorial effects (see also our newly included data for the SLC4A1 locus) would complicate our attempts to define TFs specific to domain-forming enhancers. Insofar as these efforts are ongoing and require additional functionally validated examples, we believe they fall outside the scope of this manuscript.

2. The authors note their data demonstrate the ever emerging complexity of enhancers. Did any of their analyses provide insight into further characterization of these active, hyperacetylated domains? Were they near poised gene promoters? Did cis-element motifs have location preference, such as at the center of H3K27ac or at the edges of H3K4me3, H3K9me3, etc., similar to Epigram.

We have investigated this question, but have not revealed any shared structure or localization of marks (TF binding, ATAC peaks, etc.) within HCDs. Meta-analysis of the peaks classified as HCDs did not show any patterns that would indicate the presence of putative enhancers, or of TSSs, at preferred locations within them. A meta-analysis of CTCF binding sites (using a CTCF ChIP-seq from MEL cells) indicated preferential localization at the edges of HCDs, but this result was not reproduced in the other datasets that we analyzed (using cell type-appropriate CTCF ChIP-seqs) and so we have not included it.

3. Comparison to Rose. Attempting to seek some granularity, it is logical to assume these HCD were at sites of Rose-identified SE. Many of the top regions identified by Rose are at sites critical for cell identify, containing genes encoding important cell identity determinants, signaling molecules, TF, etc. How did the author’s technique

compare, both at the specific region level, e.g. around GATA1 or KLF1 gene loci for example and at a broad scale?

We are unfortunately not completely certain what is being asked here. As we have noted, while there is significant overlap between peaks defined by our methodology as HCDs and SEs as identified by the Rose algorithm, there are a number of peaks/enhancers that are unique to each classification. Our comparison of the full HCD and SE populations in 4 cell/tissue types, as well as comparisons of the peaks called uniquely as HCDs or as SEs, indicates that HCDs are somewhat better at identifying highly-expressed and especially cell type-specific genes.

4. Comparison to Stretch E. With the various histone marks the authors downloaded (and are there more?), did the authors attempt to construct a ChromHMM-like dataset for comparison to their technique? It would be helpful to compare the two techniques.

In our opinion, for such a comparison to be generally useful, ChIP-seq tracks for multiple histone marks in addition to the two we use to define HCDs would need to be available for all of the tissue types we analyze. Outside of the ENCODE datasets employed in the stretch enhancer study, this is not generally the case. We have therefore not attempted to do such a wider analysis. In the text we acknowledge the similarities between HCDs and stretch enhancers.

5. How cell type-specific were HCD?

Any “measurement” of cell type-specificity is necessarily inexact. In relative terms the revised manuscript compares cell type-specific genes and pathways that are identified via HCD vs. SE, and now also HCD vs. broad H3K4Me3 peaks (see Major Point #4 of Reviewer #1, above). These comparisons indicate that for the cell/tissue types we have tested, HCDs are better at identifying cell type-specific genes. In a quantitative sense, however, among 216 HCDs in murine fetal liver, 441 in intestinal epithelium, and 180 in retina, we find only 12 in common between erythroid and intestinal epithelium, 6 between erythroid and retina, and 9 between intestinal epithelium and retina. (Overlap among the SE populations was similar). This might also provide a measure of specificity.

6. Were HCD aligned with other structural elements, CTCF, known TADs, etc. ?

Please see our answer to Specific Comment #2, above. We have not performed a general mapping of HCDs with respect to TADs, but HCDs are smaller (10-40 kb in murine erythroid cells, for example) than TADs and represent subregions within them. This would be consistent with our conclusion that HCDs represent a function of some enhancers, which in turn are generally not thought to act outside of their TADs.

Reviewers' comments:

Reviewer #1 (Remarks to the Author):

Overall, this paper attempts to accomplish two goals. (1) To establish the identification of HCDs as a useful method for stratifying enhancers in different tissue types and (2) to show that HCDs are functionally distinct in terms of their effect on chromatin marks compared to non-HCDs. Although it remains to be seen how general the observation of (2) is, the new data on the SLC4A1 locus is interesting and supports the assertion that at least some HCDs differ from non-HCDs as being important seed regions for maintaining H3K4me2 and H3K27ac levels. Also, even though it was a minor point, the inclusion of the PCR primers in Figure 1 clarify how well the ChIP Q-PCR data covers each locus and it makes Figures 4-7 easier to understand.

One of the underlying issues I had with the paper before was understanding how useful HCD analysis would be for the field as a whole, in other words goal (1). I am still not convinced that calling HCDs is a robust approach that could be used in multiple scenarios, although the new analysis comparing HCDs with H3K4me3 broad domains is quite intriguing.

I think that some further analysis of published datasets is necessary here to determine how robust HCD analysis is in order to support the claim that this can be easily used by other labs or in other tissues. I still think the H3K4me2 ChIP-seq pattern the authors observe in their data is unusual and not often seen, but I could be wrong about this. In addition, if the authors claim that any published ChIP-seq datasets don't meet their QC requirements, they need to be more explicit about what their metric for quality is.

Major points:

1. I think the authors misunderstood one of my requests. I would like to see some example H3K27ac, H3K4me2 and H3K4me3 tracks of HCDs, non-HCDs and SEs (similar to those shown in Figure 1) for the human erythroid data analysed for Figure 3, the murine intestinal epithelial data analysed for Figure S3 and the murine retinal cell data analysed for Figure S4. It would also be helpful if the authors could provide these analyses (along with analysis of their own data) as a UCSC session (with SEs and HCDs marked), so one can browse the data for other examples of these enhancer types. This is straight forward and often done in the review process.

2. H3K4me2 antibody specificity is an issue that was not properly addressed in the rebuttal. I initially listed this as a minor point but upon reflection, this is a key issue underlying how reproducible HCD analysis might be. It is likely that most H3K4me2 specific antibodies at least partially recognize H3K4me3, as this has been an endemic problem in the chromatin field for a very long time, especially in ChIP-seq. According to the methods in the paper, the authors use H3K4me2 (Millipore #07-030) for ChIP-seq (not an Active Motif antibody as they claim in the rebuttal). For the H3K4me2 millipore antibody, the authors don't list a lot # used, but a brief survey of histone antibody specificity databases (<http://www.histoneantibodies.com/> and <http://compbio.med.harvard.edu/antibodies/>) suggests that although the different Millipore #07-030 lots have a preference for H3K4me2, they also tend to cross-react with H3K4me3. If instead Active Motif H3K4me2 antibodies were used, they appear to be somewhat clean in dot blots from the company website (although one antibody appears to cross react with H3K4me3). However, the histoneantibodies website linked above suggests that these too have some cross-reactivity. Thus, the H3K4me2 ChIP-seq tracks could very well be a combination of H3K4me2/3 signal together (or even include H3K4me1). From a practical perspective, this is perhaps not necessarily a major problem for this paper, so long as the approach of calling HCDs is reproducible using different H3K4me2 antibodies. To address this issue, I would want to see an

analysis of HCDs in a tissue using a different ChIP-seq dataset with different antibodies than those used by the authors. The erythroid human HCDs in Figure 3 and murine intestinal epithelial HCDs (Figure S3) were identified using the same H3K4me2 Millipore antibody (07-030) as used in the current study, while the retinal data (Figure S4) appeared to use an Abcam antibody (ab7766). This provides at least one dataset using a different H3K4me2 antibody to identify HCDs. It would be ideal if at least one more dataset, not using the Millipore antibody, ideally in erythroid cells could be analysed for HCDs and compared to the current study (several datasets are available from the Vision project <http://www.bx.psu.edu/~giardine/vision/>). Although this additional analysis will not directly address the issue of specificity, from a practical perspective it will at least suggest that HCD analysis is not a product of just the Millipore (or is it Active Motif?) antibody alone, and actually represents something more robust that could be used by multiple labs using different antibodies.

3. The H3K4me3 broad domain result comparison with HCDs is interesting. However, related to point 1 above, if the H3K4me2 antibodies cross react with H3K4me3, this raises the possibility that calling H3K4me3/H3K27ac broad peaks for HCD analysis could be as effective as using H3K4me2/H3K27ac. Considering that it is likely that most if not all H3K4me2 antibodies cross react with H3K4me3 (see point 1 above), it would be useful to know if H3K4me3 could be used in place of H3K4me2 ChIP-seq for calling HCDs, or if H3K4me2 ChIP-seq is still the superior dataset for HCD analysis. I think the answer would be useful either way. It would be useful to see this analysis from at least two datasets: for example the authors own data and the H3K4me3/H3K27ac data from Hay et al, Nat Gen 2016 doi: 10.1038/ng.3605, and/or an analysis from one of the datasets from Figure 3, S4 or S3.

4. Finally, I'm not convinced that the inclusion of H3K4 methylation ChIP-seq data is necessary for the identification of HCDs. What would happen to the analysis if H3K27ac ChIP-seq alone is used? How does this compare to including H3K4me2 in the analysis? It would be useful to see this analysis using H3K27ac only using two different datasets of the authors choice.

Minor:

1. For the ChIP Q-PCR experiments in Figure 4, 5 and S6 the authors should indicate how many replicates were used and whether they were technical or biological replicates. Also, the meaning of the error bars should be indicated in the legends (e.g. SEM or SD of n biological or technical replicates).

Reviewer #2 (Remarks to the Author):

In the revised manuscript "Hyperacetylated Chromatin Domains Mark Cell Type-Specific Genes and Suggest Distinct Modes of Enhancer Function" Fox et al have responded in detail to my queries. Specifically, they have addressed the question of relationships between TF and HCD, TADs and HCD, and other structural elements with HCD in a series of new analyses. The lack of tissue or cell type specificity is still a bit puzzling.

The addition of an additional example of a domain forming HCD, adding data from the SLC4a1 locus to the Gypa region in the initial submission provides additional support for the conclusion that HCDs, at least in some cases, mediate formation of HCDs.

As noted in specific responses to Reviewer #1's comments (see below), we now include HCD analyses using H3K27Ac alone and the combination of H3K27Ac/H3K4Me3. These new analyses, along with additional consideration of potential H3K4Me antibody cross-reactivity, have resulted in the addition of a new paragraph to both the Results and Discussions sections, which we indicate as red-colored text in this revision.

Regarding H3K4Me2 antibodies, again, please refer to our response to Reviewer 1, below. We unfortunately have to report an error in our original description of the ChIP-seqs we performed using mouse fetal liver; the result of this correction, however, is that our original slate of 4 sets of ChIP-seq data actually represent 2 examples each of the 2 most commonly used H3K4Me2 antibodies in the literature - by our estimate, and also by a more careful survey in Reference 31, ~90% of publicly available H3K4Me2 ChIP-seqs.

Reviewer #1 (Remarks to the Author):

Overall, this paper attempts to accomplish two goals. (1) To establish the identification of HCDs as a useful method for stratifying enhancers in different tissue types and (2) to show that HCDs are functionally distinct in terms of their effect on chromatin marks compared to non-HCDs. Although it remains to be seen how general the observation of (2) is, the new data on the SLC4A1 locus is interesting and supports the assertion that at least some HCDs differ from non-HCDs as being important seed regions for maintaining H3K4me2 and H3K27ac levels. Also, even though it was a minor point, the inclusion of the PCR primers in Figure 1 clarify how well the ChIP Q-PCR data covers each locus and it makes Figures 4-7 easier to understand.

One of the underlying issues I had with the paper before was understanding how useful HCD analysis would be for the field as a whole, in other words goal (1). I am still not convinced that calling HCDs is a robust approach that could be used in multiple scenarios, although the new analysis comparing HCDs with H3K4me3 broad domains is quite intriguing.

I think that some further analysis of published datasets is necessary here to determine how robust HCD analysis is in order to support the claim that this can be easily used by other labs or in other tissues. I still think the H3K4me2 ChIP-seq pattern the authors observe in their data is unusual and not often seen, but I could be wrong about this. In addition, if the authors claim that any published ChIP-seq datasets don't meet their QC requirements, they need to be more explicit about what their metric for quality is.

Major points:

1. I think the authors misunderstood one of my requests. I would like to see some example H3K27ac, H3K4me2 and H3K4me3 tracks of HCDs, non-HCDs and SEs (similar to those shown in Figure 1) for the human erythroid data analysed for Figure 3, the murine intestinal epithelial data analysed for Figure S3 and the murine retinal cell data analysed for Figure S4. It would also be helpful if the authors could provide these analyses (along with analysis of their own data) as a UCSC session (with SEs and HCDs marked), so one can browse the data for other examples of these enhancer types. This is straight forward and often done in the review process.

In response to these requests, we now include example tracks, as for murine fetal liver in Fig. 1, for the human erythroid (Supplementary Fig. 2), mouse intestinal epithelial (Supplementary Fig. 3) and mouse retinal (Supplementary Fig. 4) datasets we have analyzed. [Note: Actually, Supplementary Fig. 2 was present in the revised manuscript, but we made the mistake of never referring to it in the text; this has been corrected.]. In addition, we have

created UCSC sessions for each of our analyses as well. These can be accessed at the following URLs:

https://genome.ucsc.edu/s/sfox02/Fox_etal_2020_MouseErythroid
https://genome.ucsc.edu/s/sfox02/fox_etal_2020_Human_Erythroid
https://genome.ucsc.edu/s/sfox02/fox_etal_2020_Mouse_Intestine
https://genome.ucsc.edu/s/sfox02/fox_etal_2020_Mouse_Retina

2. H3K4me2 antibody specificity is an issue that was not properly addressed in the rebuttal. I initially listed this as a minor point but upon reflection, this is a key issue underlying how reproducible HCD analysis might be. It is likely that most H3K4me2 specific antibodies at least partially recognize H3K4me3, as this has been an endemic problem in the chromatin field for a very long time, especially in ChIP-seq. According to the methods in the paper, the authors use H3K4Me2 (Millipore #07-030) for ChIP-seq (not an Active Motif antibody as they claim in the rebuttal). For the H3K4me2 millipore antibody, the authors don't list a lot # used, but a brief survey of histone antibody specificity databases (<http://www.histoneantibodies.com/> and <http://compbio.med.harvard.edu/antibodies/>) suggests that although the different Millipore #07-030 lots have a preference for H3K4me2, they also tend to cross-react with H3K4me3. If instead Active Motif H3K4me2 antibodies were used, they appear to be somewhat clean in dot blots from the company website (although one antibody appears to cross react with H3K4me3). However, the histone antibodies website linked above suggests that these too have some cross-reactivity. Thus, the H3K4me2 ChIP-seq tracks could very well be a combination of H3K4me2/3 signal together (or even include H3K4me1). From a practical perspective, this is perhaps not necessarily a major problem for this paper, so long as the approach of calling HCDs is reproducible using different H3K4me2 antibodies. To address this issue, I would want to see an analysis of HCDs in a tissue using a different ChIP-seq dataset with different antibodies than those used by the authors. The erythroid human HCDs in Figure 3 and murine intestinal epithelial HCDs (Figure S3) were identified using the same H3K4me2 Millipore antibody (07-030) as used in the current study, while the retinal data (Figure S4) appeared to use an Abcam antibody (ab7766). This provides at least one dataset using a different H3K4me2 antibody to identify HCDs. It would be ideal if at least one more dataset, not using the Millipore antibody, ideally in erythroid cells could be analysed for HCDs and compared to the current study (several datasets are available from the Vision project <http://www.bx.psu.edu/~giardine/vision/>). Although this additional analysis will not directly address the issue of specificity, from a practical perspective it will at least suggest that HCD analysis is not a product of just the Millipore (or is it Active Motif?) antibody alone, and actually represents something more robust that could be used by multiple labs using different antibodies.

Our response to this comment has resulted in a new paragraph in the Discussion (noted as red-colored text for easy reference), to address these issues. Notably, however, this comment prompted us to take a much closer look at the datasets we used, including our own, which has revealed an unfortunate mistake in our reporting in the original manuscript. The H3K4Me2 ChIP-seqs we performed using murine fetal liver actually used the same Abcam antibody (ab7766) as with the retinal ChIP-seqs. Our subsequent qPCR-ChIP analyses, however, used the Millipore antibody as previously reported. Thus, the original slate of datasets that we analyzed actually represented an even split between these two commercial products (Abcam and Millipore), which by our own estimate, and by the survey of available datasets in Reference 31 (see below), appear to have been used for ~90% of publicly available H3K4Me2 ChIP-seqs. We apologize for this mistake; we have carefully combed through notebooks to ensure that no similar confusion applies to the other antibodies we use in this study.

To address the major point, however, we agree with the reviewer that the issue of antibody specificity is an important and too often neglected one, and we appreciate the reviewer bringing this to our attention. We would go beyond the studies/controls that this reviewer discusses, however, and call attention to a 2018 publication that systematically examined H3K4Me antibodies using a technique termed “iCE-ChIP”, which involves spiking ChIP assays with semisynthetic nucleosomes with known modification patterns as controls (Reference 31 in this latest revision). Importantly, this study suggests that specificity as determined in dot-blot assays does not necessarily track with specificity as indicated by iCE-ChIP. Based on this assay, however, the commercial H3K4Me2 antibody we use for ChIP-seq (ab7766) appears to have significant cross-reactivity with H3K4Me1 but not H3K4Me3. In addition, the H3K4Me3 antibody exhibits significant cross-reactivity with H3K4Me2, while the H3K4Me1 antibody appears to be highly specific. Insofar as these are polyclonal antisera, however, none of this is necessarily applicable to our experiments, given the use of different lot numbers (although, as noted in our response to the first round of review, we have not noted any differences in results from qPCR-ChIP analyses of our commonly-assayed gene loci over the course of many years; moreover, the H3K4Me2 pattern that we observe does not correspond with the H3K4Me1 pattern very well in the ChIP-seqs that we present in this study – compare to H3K4Me3 antibodies that cross-react with H3K4Me2 in Figure 4B of Reference 31). We do not currently have the resources to perform these sorts of controls ourselves with new ChIP-seqs. As such we now include a paragraph in the Discussion that mentions these issues, and notes that the possibility of cross-reactivity among H3K4Me forms means that we cannot make conclusions regarding the specific role of H3K4Me2 in enhancer or other function.

3. The H3K4me3 broad domain result comparison with HCDs is interesting. However, related to point 1 above, if the H3K4me2 antibodies cross react with H3K4me3, this raises the possibility that calling H3K4me3/H3K27ac broad peaks for HCD analysis could be as effective as using H3K4me2/H3K27ac. Considering that it is likely that most if not all H3K4me2 antibodies cross react with H3K4me3 (see point 1 above), it would be useful to know if H3K4me3 could be used in place of H3K4me2 ChIP-seq for calling HCDs, or if H3K4me2 ChIP-seq is still the superior dataset for HCD analysis. I think the answer would be useful either way. It would be useful to see this analysis from at least two datasets: for example the authors own data and the H3K4me3/H3K27ac data from Hay et al, Nat Gen 2016 doi: 10.1038/ng.3605, and/or an analysis from one of the datasets from Figure 3, S4 or S3.

4. Finally, I’m not convinced that the inclusion of H3K4 methylation ChIP-seq data is necessary for the identification of HCDs. What would happen to the analysis if H3K27ac ChIP-seq alone is used? How does this compare to including H3K4me2 in the analysis? It would be useful to see this analysis using H3K27ac only using two different datasets of the authors choice.

In response to these 2 comments, we now include analysis of HCDs called using H3K27Ac alone, as well as the combination of H3K27Ac/H3K4Me3. Neither of these is fully as effective – as measured by average expression level of associated genes and by ability to identify erythroid-associated processes by gene ontology – as the H3K27Ac/H3K4Me2 method we originally presented. Specifically, H3K27Ac alone results in a set of associated genes that are still highly erythroid-specific, but not expressed as highly, and in fact not expressed significantly higher than SE-associated genes; while H3K27Ac/H3K4Me3 results in a set of genes perhaps slightly more highly expressed than with H3K27Ac/H3K4Me2-defined HCDs, but association of this set of genes with appropriate cell types in the Mouse Gene Atlas is inferior, and in fact (again) fails to outperform SE-associated genes. Thus, these other strategies produce overlapping but distinct sets of associated genes that still generally

outperform SE-associated genes, but not by each measure as with H3K27Ac/H3K4Me2-defined HCDs. We have added a new paragraph describing these analyses (colored as red text) to the Results section, corresponding to current Supplementary Figs. 8 and 9.

Minor:

1. For the ChIP Q-PCR experiments in Figure 4, 5 and S6 the authors should indicate how many replicates were used and whether they were technical or biological replicates. Also, the meaning of the error bars should be indicated in the legends (e.g. SEM or SD of n biological or technical replicates).

These are now provided in the Materials and Methods; we thank the reviewer for pointing out these omissions.

Reviewer #2 (Remarks to the Author):

In the revised manuscript "Hyperacetylated Chromatin Domains Mark Cell Type-Specific Genes and Suggest Distinct Modes of Enhancer Function" Fox et al have responded in detail to my queries. Specifically, they have addressed the question of relationships between TF and HCD, TADs and HCD, and other structural elements with HCD in a series of new analyses. The lack of tissue or cell type specificity is still a bit puzzling.

The addition of an additional example of a domain forming HCD, adding data from the SLC4a1 locus to the Gypa region in the initial submission provides additional support for the conclusion that HCDs, at least in some cases, mediate formation of HCDs.

REVIEWERS' COMMENTS:

Reviewer #1 (Remarks to the Author):

This paper is really excellent and I very much enjoyed reading this final version. I find the data quite complete and convincing, and I appreciate how carefully and thoughtfully the authors have addressed my concerns. I am also now quite confident that calling HCDs is an approach that will be very useful for the field as a whole for identifying key enhancers.

One very minor point for the authors: in Figure 4B and C, it might look slightly better if the bar graph data was in the same order as in 4A (ie. Wild type; +6.4 En; Pr rather than Wild type; Pr; +6.4 En), as is done in other figures with similar data as this. This is not meant to be an actual criticism that needs to be addressed, just something for the authors to consider.

Response to Reviewers:

We thank Reviewer #1 for the most recent round of comments. Reviewer #1 made one final suggestion:

“One very minor point for the authors: in Figure 4B and C, it might look slightly better if the bar graph data was in the same order as in 4A (ie. Wild type; +6.4 En; Pr rather than Wild type; Pr; +6.4 En), as is done in other figures with similar data as this. This is not meant to be an actual criticism that needs to be addressed, just something for the authors to consider.”

We actually went back and forth on this form of data presentation vs. the one we actually use in Fig. 4 several times. We feel that our current presentation is clearer about the presence of the promoter deletion in one set of cell lines; this is unique to the analysis of the GYPA gene locus, and we feel that the other way of presenting the data confuses this important point. Insofar as Reviewer 1 couched this comment as a suggestion only, in this case we would prefer to stay with the figure in its current form.